# Healthy dietary patterns and the risk of individual chronic diseases in community-dwelling adults

Xianwen Shang [1,2,3,4] ✉, Jiahao Liu[3], Zhuoting Zhu[1,2,3], Xueli Zhang[1,5], Yu Huang [1,2], Shunming Liu[1], Wei Wang [6], Xiayin Zhang[1,2], Shulin Tang[1], Yijun Hu [1], Honghua Yu [1] ✉, Zongyuan Ge[7] & Mingguang He [1,3,8,9,10] ✉

It is unclear regarding associations of dietary patterns with a wide range of chronic diseases and which dietary score is more predictive of major chronic diseases. Using the UK Biobank, we examine associations of four individual healthy dietary scores with the risk of 48 individual chronic diseases. Higher Alternate Mediterranean Diet score is associated with a lower risk of 32 (all 8 cardiometabolic disorders, 3 out of 10 types of cancers, 7 out of 10 psychological/neurological disorders, 5 out of 6 digestive disorders, and 9 out of 14 other chronic diseases). Alternate Healthy Eating Index-2010 and Healthful Plant-based Diet Index are inversely associated with the risk of 29 and 23 individual chronic diseases, respectively. A higher Anti-Empirical Dietary Inflammatory Index is associated with a lower risk of 14 individual chronic diseases and a higher incidence of two diseases. Our findings support dietary guidelines for the prevention of most chronic diseases.

It was estimated that the global population of individuals aged 65 years and above was 727 million in 2020 and this number is anticipated to rise to 1.5 billion by the year 2050[1]. Meanwhile, ageing is one of the most important risk factors for the development of noncommunicable diseases (NCDs), including cardiovascular disease (CVD), diabetes, cancers, and neurodegenerative diseases[2,3]. As the percentage of people aged 65 years and older grew from 6.1% in 1990 to 8.8% in 2017, this demographic shift was linked to an additional 12 million global deaths[4]. The total number of deaths caused by NCDs increased by 22.7% from 2007 to 2017 and NCDs accounted for 73.4% (41.1 million) of total deaths in 2017[5]. The increasing prevalence of these chronic diseases within the global

aging population imposes a significant burden on both the economic and healthcare systems. Therefore, identifying intervention priorities that promote healthy ageing and prevent or delay the development of chronic conditions is of paramount importance. In addition to these age-related chronic diseases, psychiatric/neurological disorders including alcohol use disorder, other psychoactive substance abuse, schizophrenia, multiple sclerosis, and migraine have been linked to elevated mortality risks[6-10]. While digestive disorders like dyspepsia, irritable bowel syndrome, and inflammatory bowel disease might not directly contribute to mortality risk, these conditions are widespread and place a substantial load on healthcare and economic systems.[11-13]. Therefore, investigating

[1]Guangdong Eye Institute, Department of Ophthalmology, Guangdong Provincial People's Hospital (Guangdong Academy of Medical Sciences), Southern Medical University, Guangzhou 510080, China. [2]Guangdong Cardiovascular Institute, Guangdong Provincial People's Hospital, Guangdong Academy of Medical Sciences, Guangzhou 510080, China. [3]Centre for Eye Research Australia, Melbourne, VIC 3002, Australia. [4]Department of Medicine (Royal Melbourne Hospital), University of Melbourne, Melbourne, VIC 3050, Australia. [5]Medical Research Institute, Guangdong Provincial People's Hospital (Guangdong Academy of Medical Sciences), Southern Medical University, Guangzhou 510080, China. [6]State Key Laboratory of Ophthalmology, Zhongshan Ophthalmic Center, Sun Yat-sen University, Guangzhou 510060, China. [7]Monash e-Research Center, Faculty of Engineering, Airdoc Research, Nvidia AI Technology Research Center,  Monash University, Melbourne, VIC 3800, Australia. [8]School of Optometry, The Hong Kong Polytechnic University, Hong Kong, China. [9]Research Centre for SHARP Vision, The Hong Kong Polytechnic University, Hong Kong, China. [10]Centre for Eye and Vision Research (CEVR), Hong Kong, China. ✉e-mail: andy243@126.com; yuhonghua@gdph.org.cn; mingguang.he@polyu.edu.hk

significant modifiable factors for these non-age-related chronic conditions also holds considerable interest.

Evidence from previous studies indicates that adopting a healthy lifestyle is linked to a decreased risk of chronic diseases and mortality, as well as an increased likelihood of experiencing healthy aging[14–17]. Diet plays a vital role as a lifestyle factor for chronic conditions and can be effectively targeted for the prevention and delay of these conditions[18–21]. Previous studies have demonstrated that adherence to some dietary patterns such as the Mediterranean diet score, Healthy Eating Index, and Dietary Approaches to Stop Hypertension was associated with a lower risk of CVDs, diabetes, and certain types of cancers[18,19,22]. It has been inconsistent regarding the association between dietary patterns and neurodegenerative diseases such as dementia and Parkinson's disease[20,23]. Many previous studies have investigated the association between dietary patterns and bone health and found that some dietary patterns might be beneficial for the risk reduction of musculoskeletal disorders[24]. However, less is known regarding the association between dietary patterns and the risk of other chronic diseases such as respiratory diseases, digestive disorders, chronic kidney disease (CKD), eczema, endocrine disorders, and ophthalmic conditions[25–27].

Although previous studies have linked dietary patterns to some major chronic conditions, whether these dietary patterns are associated with the risk of a wide range of individual chronic diseases remains to be explored. Using the UK Biobank, we aimed to associate four commonly used dietary scores with the risk of a wide range of individual chronic diseases and examine which dietary pattern was more predictive of chronic diseases.

## Results

### Population selection
Individuals with no data on diet ($n = 295,101$), or with only one dietary assessment ($n = 83,413$) were excluded from the analysis. In addition, individuals with total energy intake in either the highest or lowest percentile were excluded from the analysis ($n = 2478$). We included 121,513 participants (55.9% females) aged 30-75 (mean ± SD: 59.0 ± 7.9) years at baseline in the final analysis.

Individuals with higher dietary scores were more likely to be older, be highly educated, exercise, and less likely to smoke. Individuals with higher Alternate Mediterranean Diet score (AMED), Alternate Healthy Eating Index-2010 (AHEI-2010), or Healthful Plant-based Diet Index (HPDI) scores were more likely to be female whereas those with a higher anti-Empirical Dietary Inflammatory Index (AEDII) score were more likely to be male (Table 1).

### Incidence of individual diseases
Due to variations in the number of participants included in the analysis for each disease, the duration of follow-up also differed across these diseases. The mean follow-up duration ranged from 8.4 years for dyspepsia to 8.6 years for multiple sclerosis. The number of newly diagnosed cases varied from 94 for multiple sclerosis to 9815 for dyspepsia.

### Dietary scores and cardiometabolic disorders
After controlling for false discovery rate (FDR), AMED, AHEI-2010, and HPDI dietary scores were inversely associated with the risk of all individual cardiometabolic disorders (CMDs). In the fully adjusted model, the associations between AMED and all CMDs remained significant (adjusted HR (95% CI) for CVD associated with each quintile increment in the dietary score: 0.94 (0.93-0.95)), hypertension (0.94 (0.92-0.96)), and diabetes (0.96 (0.93-0.99)). Out of the four dietary scores, the AMED score yielded the lowest hazard ratios (HRs) for various CMDs except for diabetes. AHEI-2020 was inversely associated with the risk of other CMDs including CVD (0.96 (0.94-0.97)) and hypertension (0.93 (0.91-0.95)) but not diabetes (0.98 (0.95-1.01)). Further analysis showed that BMI attenuated the association between AHEI-2010 and

incident diabetes to be non-significant. After adjustment for all covariates, HPDI was significantly associated with the risk of 5 CMDs. AEDII was inversely associated with the risk of 6 CMDs in the full model. Particularly, AEDII had the largest effect size for the prevention of diabetes (HR (95% CI): 0.89 (0.86-0.92)) (Fig. 1). Similar results were seen when dietary scores were analyzed as categorical variables (Tables S1–S4).

### Dietary scores and cancers
After controlling for FDR, AMED (HR (95% CI) for each quintile increment: 0.93 (0.91-0.95)), AHEI-2010: (0.95 (0.93-0.97)), and HPDI: (0.95 (0.93-0.97)) but not AEDII (1.00 (0.98-1.02)) dietary scores were inversely associated with the risk of all cancers with AMED yielding the lowest risk. For types of cancer, a higher AMED score was associated with a lower risk of lung cancer, oesophageal cancer, and other cancers in the full model. Higher AHEI-2010 was associated with a lower risk of non-melanoma skin cancer, lung cancer, breast cancer, and other cancers. Higher HPDI was associated with a lower risk of colon cancer, ovarian cancer, and other cancers (Fig. 2, Tables S1–S4).

### Dietary scores and psychological/neurological disorders
AMED and AHEI-2010 but not other dietary scores were inversely associated with the risk of dementia (HR (95% CI) for each quintile increment in AMED: 0.92 (0.87-0.98); AHEI-2010: 0.93 (0.88-0.98)). Higher AMED was also associated with a lower risk of Parkinson's disease (HR (95% CI): 0.92 (0.86-0.99)). All four dietary scores were inversely associated with a reduced risk of depression. AMED and AHEI-2010 scores were inversely but the AEDII score was positively associated with the risk of alcohol use disorder and psychoactive substance abuse. AMED, AHEI-2010, and HPDI dietary scores were inversely associated with the risk of epilepsy. AMED yielded the lowest risk for dementia, Parkinson's disease, depression, anxiety, and epilepsy, whilst AHEI-2010 yielded the lowest risk for alcohol use disorder and psychoactive substance abuse (Fig. 3, Tables S1–S4).

### Dietary scores and digestive disorders
All four dietary scores were inversely associated with the risk of dyspepsia, treated constipation, diverticular disease, irritable bowel syndrome, and chronic liver disease. AMED yielded the lowest risk for treated constipation and chronic liver disease (Fig. 4, Tables S1–S4).

### Dietary scores and other chronic diseases
All four dietary scores were inversely associated with the risk of chronic obstructive pulmonary disease ([COPD], HR (95% CI) for AMED: 0.88 (0.85-0.91), AEDII: 0.92 (0.89-0.95), AHEI-2010: 0.90 (0.87-0.93), HPDI: 0.92 (0.89-0.95), CKD (AMED: 0.89 (0.87-0.91), AEDII: 0.91 (0.89-0.93), AHEI-2010: 0.94 (0.92-0.96), HPDI: 0.91 (0.89-0.93)), and prostate disorders (AMED: 0.97 (0.94-0.99), AEDII: 0.97 (0.95-0.995), AHEI-2010: 0.97 (0.95-0.995), HPDI: 0.96 (0.94-0.99)). A higher AMED score was also associated with a lower risk of asthma, bronchiectasis, eczema, cataract, and pernicious anemia. AMED yielded the lowest risk for COPD, bronchiectasis, CKD, and pernicious anemia, whist only AEDII was inversely associated with the risk of thyroid disorders (Fig. 5, Tables S1–S4).

### AMED components and chronic diseases
Out of 450 associations, 155 were significant after adjustment for FDR with 154 being inverse associations. Recommended intakes of whole grains, vegetables, fruits, nuts, legumes, fish, monounsaturated fat to saturated fat ratio, and alcohol were associated with a lower risk of most chronic diseases and not significantly associated with a higher risk of any chronic diseases. Recommended intake (low level) of red meat was associated with a lower risk of diabetes, CKD, diverticular disease, and osteoporosis, but a higher risk of Meniere's disease (Fig. 6).

**Table 1 | Baseline characteristics across quintiles of dietary scores**

|  | AMED | | AEDII | | AHEI-2010 | | HPDI | |
|---|---|---|---|---|---|---|---|---|
|  | Quintile 1 | Quintile 5 | Quintile 1 | Quintile 5 | Quintile 1 | Quintile 5 | Quintile 1 | Quintile 5 |
| Age (years) | 57.8 ± 8.1 | 60.0 ± 7.6 | 57.5 ± 8.1 | 60.1 ± 7.5 | 58.1 ± 8.1 | 59.6 ± 7.7 | 58.1 ± 8.2 | 59.5 ± 7.5 |
| Ethnicity |  |  |  |  |  |  |  |  |
| White | 22307 (97.2) | 27408 (96.0) | 22855 (94.0) | 23924 (98.4) | 23601 (97.1) | 23182 (95.4) | 29113 (96.6) | 23535 (96.5) |
| Non-white | 640 (2.8) | 1146 (4.0) | 1448 (6.0) | 379 (1.6) | 702 (2.9) | 1121 (4.6) | 1025 (3.4) | 845 (3.5) |
| Sex (female) | 10911 (47.5) | 18162 (63.6) | 13629 (56.1) | 13237 (54.5) | 11535 (47.5) | 15267 (62.8) | 12939 (42.9) | 16833 (69.0) |
| Education* |  |  |  |  |  |  |  |  |
| Low | 1964 (8.6) | 1374 (4.8) | 2034 (8.4) | 1248 (5.1) | 1705 (7.0) | 1590 (6.5) | 2340 (7.8) | 1249 (5.1) |
| Intermediate | 11959 (52.1) | 11657 (40.8) | 12780 (52.6) | 10050 (41.4) | 11999 (49.4) | 10831 (44.6) | 15263 (50.6) | 10111 (41.5) |
| High | 9024 (39.3) | 15523 (54.4) | 9489 (39.0) | 13005 (53.5) | 10599 (43.6) | 11882 (48.9) | 12535 (41.6) | 13020 (53.4) |
| Household income (pounds) |  |  |  |  |  |  |  |  |
| <18,000 | 3266 (14.2) | 3353 (11.7) | 3740 (15.4) | 2484 (10.2) | 2856 (11.8) | 3628 (14.9) | 3961 (13.1) | 2962 (12.1) |
| 18,000-30,999 | 6976 (30.4) | 8834 (30.9) | 7792 (32.1) | 7081 (29.1) | 6897 (28.4) | 8016 (33.0) | 9161 (30.4) | 7437 (30.5) |
| 31,000-51,999 | 6079 (26.5) | 7541 (26.4) | 6327 (26.0) | 6437 (26.5) | 6520 (26.8) | 6312 (26.0) | 8137 (27.0) | 6333 (26.0) |
| 52,000-100,000 | 5110 (22.3) | 6687 (23.4) | 5159 (21.2) | 6114 (25.2) | 6053 (24.9) | 4946 (20.4) | 6931 (23.0) | 5775 (23.7) |
| >100,000 | 1516 (6.6) | 2139 (7.5) | 1285 (5.3) | 2187 (9.0) | 1977 (8.1) | 1401 (5.8) | 1948 (6.5) | 1873 (7.7) |
| Current alcohol drinkers |  |  |  |  |  |  |  |  |
| Yes | 21455 (93.5) | 27119 (95.0) | 21540 (88.6) | 23996 (98.7) | 23768 (97.8) | 21632 (89.0) | 28394 (94.2) | 22825 (93.6) |
| No | 1492 (6.5) | 1435 (5) | 2763 (11.4) | 307 (1.2) | 535 (2.2) | 2671 (10.9) | 1744 (5.8) | 1555 (6.4) |
| Smoking |  |  |  |  |  |  |  |  |
| Never | 12390 (54.0) | 17150 (60.1) | 15294 (62.9) | 12034 (49.5) | 12760 (52.5) | 14814 (61.0) | 17209 (57.1) | 13969 (57.3) |
| Former | 8086 (35.2) | 10173 (35.6) | 7427 (30.6) | 10200 (42.0) | 9021 (37.1) | 8309 (34.2) | 10289 (34.1) | 9104 (37.3) |
| Current | 2471 (10.8) | 1231 (4.3) | 1582 (6.5) | 2069 (8.5) | 2522 (10.4) | 1180 (4.9) | 2640 (8.8) | 1307 (5.4) |
| Physical activity (MET-minutes/week) | 2259 ± 2165 | 2602 ± 2206 | 2419 ± 2266 | 2456 ± 2123 | 2252 ± 2075 | 2671 ± 2307 | 2290 ± 2169 | 2638 ± 2225 |
| Sleep duration (hours) | 7.2 ± 1.1 | 7.2 ± 0.9 | 7.2 ± 1.0 | 7.2 ± 0.9 | 7.2 ± 1.0 | 7.2 ± 1.0 | 7.2 ± 1.0 | 7.2 ± 1.0 |
| Total energy intake (KJ/day) | 8281 ± 2075 | 9237 ± 2028 | 8889 ± 2138 | 8874. ± 2039 | 8680 ± 2065 | 8955 ± 2126 | 9565 ± 2103 | 8025 ± 1849 |
| Genetic score for longevity† | 0.49 ± 0.05 | 0.49 ± 0.05 | 0.49 ± 0.05 | 0.49 ± 0.05 | 0.49 ± 0.05 | 0.49 ± 0.05 | 0.49 ± 0.05 | 0.49 ± 0.05 |

*AEDII* Anti-Empirical Dietary Inflammatory Index, *AHEI-2010* Alternate Healthy Eating Index-2010, *AMED* Alternate Mediterranean Diet score, *HPDI* Healthful Plant-based Diet Index, *MET* metabolic equivalent.
Data are means ± standard deviations, or N (%).
*Educational levels were classified as low for 0-5 years, intermediate for 6-12 years, and high for ≥13 years.
†Genetic risk score was calculated for longevity was calculated using 78 single-nucleotide polymorphisms.

## Moderation analysis

The inverse association between AMED and the incidence of irritable bowel syndrome, osteoporosis, dyspepsia, and cataract was stronger among individuals with hypertension/dyslipidemia (Fig. S1). An AEDII score was more predictive of diabetes/CKD among younger than in older individuals (Fig. S2). Age and metabolic disorders were significant moderators for the association between AHEI-2010 and the risk of cancer or cataract. The inverse association between AHEI-2010 and incident dyspepsia was stronger among individuals with lower education (Fig. S3). The association between HPDI score and incident hypertension was stronger in younger than older individuals (Fig. S4).

## Sensitivity analysis

Inverse associations between AMED, AHEI-2010, and HPDI dietary scores and the incidence of CMDs, cancers, psychological/neurological disorders, digestive disorders, and most other conditions were observed when the analysis was conducted among individuals by excluding those developed in the first 4 years of follow-up. Higher AEDII was associated with a lower risk of CVD, diabetes, digestive disorders, CKD, COPD, osteoporosis, pernicious anemia, and eczema and a higher risk of alcohol use disorder and psychoactive substance abuse (Figs. S5–S9, Tables S5–S8). Similar results for the association between dietary scores and the risk of individual diseases were seen

among individuals with ≥3 dietary assessments compared with the primary analyses (Figs. S10–S14, Tables S9–S12).

## Discussion

In this large cohort study, we found a higher AMED score was associated with a lower risk of 32 (all 8 CMDs, 3 out of 10 types of cancers, 7 out of 10 psychological/neurological disorders, 5 out of 6 digestive disorders, and 9 out of 14 other chronic diseases) out of 48 chronic diseases. AHEI-2010 was inversely associated with the risk of 29 chronic diseases (7 CMDs, 4 cancers, 5 psychological/neurological disorders, 5 digestive disorders, and 8 other chronic diseases). A higher HPDI score was associated with a reduced risk of 23 chronic diseases (6 CMDs, 4 cancers, 4 psychological/neurological disorders, 5 digestive disorders, and 4 other chronic diseases). No positive associations between AMED, AHEI-2010, and HPDI and the risk of any chronic disease were observed. AEDII was inversely associated with the risk of 14 chronic diseases and positively associated with the risk of two chronic conditions (alcohol use disorder, psychoactive substance abuse). AHEI-2010 demonstrated the lowest risk for alcohol use disorder and psychoactive substance abuse, AEDII showed the lowest risk for diabetes and thyroid disorders, while AMED yielded the lowest risk for many other chronic diseases (CVD, cancer, COPD, CKD, chronic liver disease, psychological/neurological disorders, and digestive

| Cardiometabolic disorder | Events/participants | Incidence | Hazard ratio (95% CI), Model 1 | Hazard ratio (95% CI), Model 2 |
|---|---|---|---|---|
| Cardiovascular disease | 8925/111746 | 9.18 | | |
| AMED | | | 0.92 (0.90-0.93)* | 0.94 (0.93-0.95)* |
| AEDII | | | 0.95 (0.93-0.96)* | 0.96 (0.95-0.98)* |
| AHEI-2010 | | | 0.94 (0.93-0.96)* | 0.96 (0.94-0.97)* |
| HPDI | | | 0.95 (0.94-0.96)* | 0.97 (0.95-0.98)* |
| Coronary heart disease | 4820/113581 | 4.88 | | |
| AMED | | | 0.92 (0.90-0.94)* | 0.95 (0.93-0.97)* |
| AEDII | | | 0.93 (0.91-0.95)* | 0.95 (0.93-0.97)* |
| AHEI-2010 | | | 0.95 (0.93-0.97)* | 0.97 (0.95-0.99)* |
| HPDI | | | 0.94 (0.92-0.96)* | 0.96 (0.94-0.98)* |
| Atrial fibrillation | 1831/114144 | 1.84 | | |
| AMED | | | 0.92 (0.89-0.95)* | 0.95 (0.92-0.98)* |
| AEDII | | | 0.97 (0.94-0.995) | 0.99 (0.96-1.03) |
| AHEI-2010 | | | 0.93 (0.90-0.96)* | 0.95 (0.92-0.98)* |
| HPDI | | | 0.96 (0.93-0.99)* | 0.99 (0.96-1.02) |
| Heart failure | 1518/114950 | 1.52 | | |
| AMED | | | 0.87 (0.84-0.90)* | 0.91 (0.88-0.95)* |
| AEDII | | | 0.91 (0.88-0.94)* | 0.95 (0.91-0.98)* |
| AHEI-2010 | | | 0.93 (0.90-0.97)* | 0.96 (0.92-0.99)* |
| HPDI | | | 0.92 (0.89-0.96)* | 0.96 (0.92-0.996)* |
| Other cardiac problem | 3586/114437 | 3.6 | | |
| AMED | | | 0.90 (0.88-0.93)* | 0.93 (0.91-0.95)* |
| AEDII | | | 0.95 (0.92-0.97)* | 0.96 (0.94-0.99)* |
| AHEI-2010 | | | 0.93 (0.90-0.95)* | 0.94 (0.92-0.96)* |
| HPDI | | | 0.95 (0.93-0.97)* | 0.97 (0.94-0.99)* |
| Stroke | 719/114846 | 0.72 | | |
| AMED | | | 0.88 (0.83-0.93)* | 0.90 (0.85-0.94)* |
| AEDII | | | 0.96 (0.91-1.01) | 0.97 (0.92-1.03) |
| AHEI-2010 | | | 0.89 (0.84-0.94)* | 0.90 (0.85-0.95)* |
| HPDI | | | 0.94 (0.89-0.99)* | 0.96 (0.91-1.01) |
| Peripheral vascular disease | 870/114860 | 0.87 | | |
| AMED | | | 0.90 (0.86-0.95)* | 0.92 (0.88-0.96)* |
| AEDII | | | 0.96 (0.92-1.01) | 0.96 (0.92-1.01) |
| AHEI-2010 | | | 0.92 (0.88-0.97)* | 0.93 (0.89-0.98)* |
| HPDI | | | 0.95 (0.90-0.99)* | 0.95 (0.91-0.99)* |
| Hypertension | 3882/88897 | 5.01 | | |
| AMED | | | 0.90 (0.88-0.92)* | 0.94 (0.92-0.96)* |
| AEDII | | | 0.94 (0.92-0.96)* | 0.98 (0.96-1.001) |
| AHEI-2010 | | | 0.91 (0.89-0.93)* | 0.93 (0.91-0.95)* |
| HPDI | | | 0.90 (0.88-0.92)* | 0.94 (0.92-0.96)* |
| Diabetes | 2451/111156 | 2.53 | | |
| AMED | | | 0.88 (0.85-0.90)* | 0.96 (0.93-0.99)* |
| AEDII | | | 0.82 (0.80-0.84)* | 0.89 (0.86-0.92)* |
| AHEI-2010 | | | 0.93 (0.91-0.96)* | 0.98 (0.95-1.01) |
| HPDI | | | 0.86 (0.84-0.89)* | 0.93 (0.90-0.96)* |

disorders). The major contributors to the benefits of AMED were higher intakes of whole grains, vegetables, fruits, nuts, legumes, and fish and lower red meat intakes.

Our findings are consistent with previous studies demonstrating that healthy dietary patterns were associated with a lower risk of CMDs including CVD, diabetes, and hypertension. Data from the Nurses' Health Study and Health Professionals Follow-up Study showed that individuals in the highest quintile of AMED (HR) (95% CI: 0.83 (0.79-0.86)), HPDI (0.86 (0.82-0.89)), and AHEI (0.79 (0.75-0.82)) dietary scores had a lower risk of CVD compared with those in the lowest quintile[28]. Data from the Atherosclerosis Risk in Communities Study also found an inverse association between AHEI-2010, AMED, and the

**Fig. 1 | The association between dietary scores and the risk of cardiometabolic disorders.** AEDII, Anti-Empirical Dietary Inflammatory Index; AHEI-2010, Alternate Healthy Eating Index-2010; AMED, Alternate Mediterranean Diet score; CI, confidence interval; HPDI, Healthful Plant-based Diet Index. The incidence refers to the number of event cases per 1000 person-years. Cardiovascular disease includes coronary heart disease, heart failure, atrial fibrillation, other cardiac disease, stroke, and peripheral vascular disease. Cox proportional hazard regression models were used to examine associations of each of the four dietary scores with the risk of individual cardiometabolic disorders adjusted for potential confounding variables.

Model 1 was adjusted for age, sex, and total energy intake; Model 2 was adjusted for Model 1 plus ethnicity, education, income, BMI, smoking, sleep, physical activity, and GRS for longevity. Dietary scores were analyzed as continuous variables (each quintile increment). The vertical dash lines represent the hazard ratio of 1. Squares represent the hazard ratios (black color for AMED, orange color for AEDII, blue color for AHEI-2010, and green color for HPDI). Horizontal lines indicate the range of the 95% confidence interval. *Indicates a significant association through two-sided statistical tests while controlling for FDR.

risk of CVD and CVD mortality[29]. Previous studies also revealed that healthy dietary patterns were associated with a reduced risk of coronary heart disease, heart failure, stroke, diabetes, and CVD risk factors[18,19]. AEDII was inversely associated with the risk of other CMDs but not stroke or peripheral vascular disease in our study. This is in line with a recent umbrella review reporting that no evidence was seen for the association between AEDII and the risk of stroke[30]. Our study suggests adherence to healthy dietary patterns especially AMED, AHEI, and HPDI may help minimize the risk of CMDs.

We found AMED, AHEI, and HPDI dietary scores were inversely associated with the risk of all cancers and some types of cancers. Our results are consistent with previous studies regarding the associations between dietary scores and the risk of all cancers[18,19]. Likely, a recent meta-analysis reported that healthy dietary scores were associated with a lower risk of lung cancer (RR (95% CI) for HEI: 0.87 (0.80–0.95); AHEI: 0.88 (0.81–0.95), Mediterranean diet: 0.87 (0.81–0.93))[31]. We found no significant association between AEDII and the incidence of lung cancer but the meta-analysis observed a positive association between DII and the risk of lung cancer (RR (95% CI): 1.14 (1.07–1.22))[31]. This might be attributable to the different study population backgrounds and the methods of the DII calculation. Our results regarding the inverse association between AMED, AHEI, and HPDI dietary scores and the risk of colon cancer are consistent with previous studies[18,19]. Only three cohort studies have reported the association between MED and the risk of oesophageal cancer with inconsistent results[32]. In our study, AMED but not other dietary scores were inversely associated with the risk of oesophageal cancer. The main contributors to the potential protective effect of AMED on oesophageal cancer were high fish and moderate alcohol intakes as shown in Fig. 6. However, this needs to be confirmed by more future research.

Our data demonstrated that only the AMED dietary score was inversely associated with the risk of dementia and/or Parkinson's disease. A pooled analysis from three large cohort studies in the USA observed an inverse association between healthy dietary scores and incident dementia[33]. A recent prospective analysis based on the UK Biobank showed that the traditional MED score created by Trichopoulou et al. was not significantly associated with the risk of dementia (HR (95% CI) 0.88 (0.70-1.11) for Tertile 3 versus Tertile 1)[34]. The different results between Zhang et al.'s study and ours may be related to the different methods of the calculation of MED score where we treated nuts as a separate food group and excluded dairy from the calculation. A systematic review reported that seven previous studies yielded mixed results on the association between dietary scores and the risk of Parkinson's disease, which might be due to differences in the mean population dietary patterns and methods of calculating dietary scores[35]. A diet rich in antioxidants and anti-inflammatory compounds, as found in the Mediterranean diet, may contribute to reducing inflammation and oxidative stress in the brain[36,37], which are risk factors for epilepsy. This may partly explain why we found an inverse association between AMED and incident epilepsy. Our study is in line with many previous studies supporting an inverse association between healthy dietary patterns and the incidence of depression and anxiety[38,39]. The positive association between AEDII and the risk of alcohol use disorder could be explained by the substantial role of alcohol consumption as a key component of AEDII. Our study supports

the idea that adherence to healthy dietary patterns, especially AMED may help promote psychological and neurological health.

Our study provides further evidence on the potential favorable effects of healthy dietary patterns on the prevention of digestive disorders. AEDII and HPDI dietary scores were inversely associated with the risk of all five gastrointestinal diseases including dyspepsia, treated constipation, diverticular disease, inflammatory bowel disease, and irritable bowel syndrome. Although the association between dietary scores and the incidence of gastrointestinal diseases has not been reported in previous studies, components of dietary indices such as whole grains, vegetables, fruits, and red meat have been linked to gastrointestinal diseases[40–43]. Evidence suggests adherence to healthy dietary patterns may help promote the composition of the microbiota and reduce inflammation and oxidative stress[44,45], which is associated with a lower risk of gastrointestinal diseases[40]. We found all four healthy dietary scores were inversely associated with a lower risk of chronic liver disease. Likely, a recent meta-analysis reported that prudent (OR) (95% CI): 0.78 (0.71–0.85) and Mediterranean (0.77 (0.60-0.98)) dietary patterns were associated with a reduced risk of non-alcoholic fatty liver disease[26]. However, few studies have reported the association between AEDII and HPDI dietary scores and the risk of chronic liver disease.

As previous studies on other chronic diseases (not mentioned above) are limited by small sample sizes or short duration of follow-up, our study provides additional evidence on the benefits of healthy dietary patterns on the prevention of these diseases. The effects of diet on the development of CKD should not be overlooked. Recent evidence suggests that plant-based dietary patterns with higher intakes of whole, unprocessed foods, preferably from plant-based sources (especially the MED diet) may help prevent and treat CKD[46]. This is consistent with our study revealing that all four healthy dietary patterns were inversely associated with the risk of CKD. The higher intakes of whole grains, vegetables, fruits, nuts, legumes, and fish and lower intakes of red/processed meat are the major contributors to the benefits of healthy dietary patterns (Fig. 6). We found dietary scores especially AMED and AHEI-2010 were associated with a reduced risk of COPD, asthma, and bronchiectasis. Although individual dietary intakes including fruits, vegetables, and whole grains have been linked to respiratory diseases, several prospective cohort studies have investigated the association between dietary indices and the risk of respiratory diseases[25,47]. Several cohort studies demonstrated that adherence to the AHEI pattern was associated with a lower risk of COPD and mortality from COPD[25]. Only two cohort studies reported the association between dietary patterns and incident asthma with inconsistent findings[47]. Our cohort study with a large sample size is the first to demonstrate an inverse association between healthy dietary scores and incident bronchiectasis, thus which needs to be warranted in future prospective cohorts. Although we found dietary scores were not significantly associated with the risk of fracture, all four dietary scores were inversely associated with a lower incidence of osteoporosis (associations with AMED and AHEI-2010 remained significant in the full model). A meta-analysis suggests that healthy dietary patterns were associated with a lower risk of fracture[47]. The inconsistent results between the meta-analysis and our study may be due to the differences in the definition of dietary patterns and facture. Some previous studies are in line with ours showing an inverse association

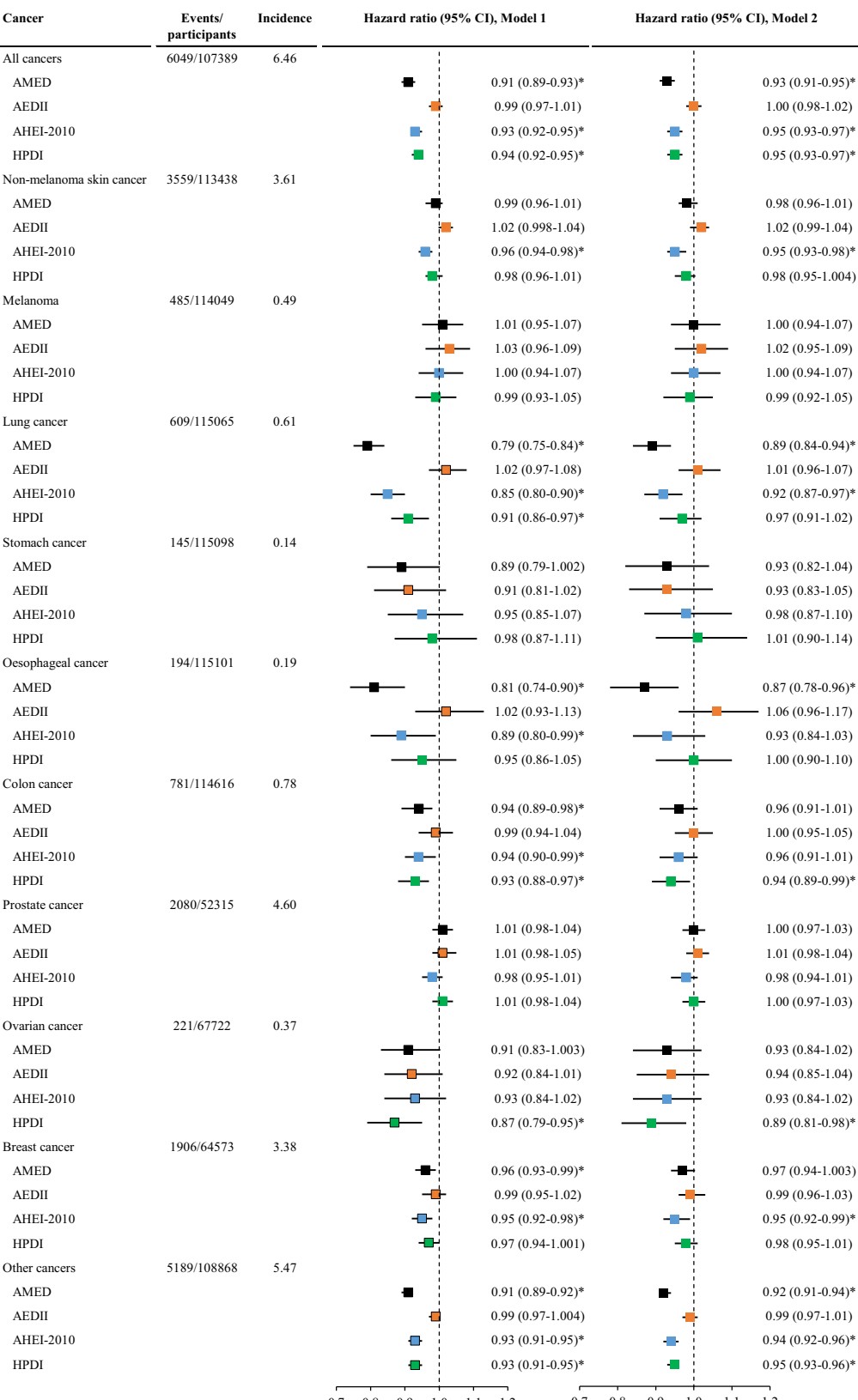

| Cancer | Events/participants | Incidence | Hazard ratio (95% CI), Model 1 | | Hazard ratio (95% CI), Model 2 | |
|---|---|---|---|---|---|---|
| All cancers | 6049/107389 | 6.46 | | | | |
| AMED | | | | 0.91 (0.89-0.93)* | | 0.93 (0.91-0.95)* |
| AEDII | | | | 0.99 (0.97-1.01) | | 1.00 (0.98-1.02) |
| AHEI-2010 | | | | 0.93 (0.92-0.95)* | | 0.95 (0.93-0.97)* |
| HPDI | | | | 0.94 (0.92-0.95)* | | 0.95 (0.93-0.97)* |
| Non-melanoma skin cancer | 3559/113438 | 3.61 | | | | |
| AMED | | | | 0.99 (0.96-1.01) | | 0.98 (0.96-1.01) |
| AEDII | | | | 1.02 (0.998-1.04) | | 1.02 (0.99-1.04) |
| AHEI-2010 | | | | 0.96 (0.94-0.98)* | | 0.95 (0.93-0.98)* |
| HPDI | | | | 0.98 (0.96-1.01) | | 0.98 (0.95-1.004) |
| Melanoma | 485/114049 | 0.49 | | | | |
| AMED | | | | 1.01 (0.95-1.07) | | 1.00 (0.94-1.07) |
| AEDII | | | | 1.03 (0.96-1.09) | | 1.02 (0.95-1.09) |
| AHEI-2010 | | | | 1.00 (0.94-1.07) | | 1.00 (0.94-1.07) |
| HPDI | | | | 0.99 (0.93-1.05) | | 0.99 (0.92-1.05) |
| Lung cancer | 609/115065 | 0.61 | | | | |
| AMED | | | | 0.79 (0.75-0.84)* | | 0.89 (0.84-0.94)* |
| AEDII | | | | 1.02 (0.97-1.08) | | 1.01 (0.96-1.07) |
| AHEI-2010 | | | | 0.85 (0.80-0.90)* | | 0.92 (0.87-0.97)* |
| HPDI | | | | 0.91 (0.86-0.97)* | | 0.97 (0.91-1.02) |
| Stomach cancer | 145/115098 | 0.14 | | | | |
| AMED | | | | 0.89 (0.79-1.002) | | 0.93 (0.82-1.04) |
| AEDII | | | | 0.91 (0.81-1.02) | | 0.93 (0.83-1.05) |
| AHEI-2010 | | | | 0.95 (0.85-1.07) | | 0.98 (0.87-1.10) |
| HPDI | | | | 0.98 (0.87-1.11) | | 1.01 (0.90-1.14) |
| Oesophageal cancer | 194/115101 | 0.19 | | | | |
| AMED | | | | 0.81 (0.74-0.90)* | | 0.87 (0.78-0.96)* |
| AEDII | | | | 1.02 (0.93-1.13) | | 1.06 (0.96-1.17) |
| AHEI-2010 | | | | 0.89 (0.80-0.99)* | | 0.93 (0.84-1.03) |
| HPDI | | | | 0.95 (0.86-1.05) | | 1.00 (0.90-1.10) |
| Colon cancer | 781/114616 | 0.78 | | | | |
| AMED | | | | 0.94 (0.89-0.98)* | | 0.96 (0.91-1.01) |
| AEDII | | | | 0.99 (0.94-1.04) | | 1.00 (0.95-1.05) |
| AHEI-2010 | | | | 0.94 (0.90-0.99)* | | 0.96 (0.91-1.01) |
| HPDI | | | | 0.93 (0.88-0.97)* | | 0.94 (0.89-0.99)* |
| Prostate cancer | 2080/52315 | 4.60 | | | | |
| AMED | | | | 1.01 (0.98-1.04) | | 1.00 (0.97-1.03) |
| AEDII | | | | 1.01 (0.98-1.05) | | 1.01 (0.98-1.04) |
| AHEI-2010 | | | | 0.98 (0.95-1.01) | | 0.98 (0.94-1.01) |
| HPDI | | | | 1.01 (0.98-1.04) | | 1.00 (0.97-1.03) |
| Ovarian cancer | 221/67722 | 0.37 | | | | |
| AMED | | | | 0.91 (0.83-1.003) | | 0.93 (0.84-1.02) |
| AEDII | | | | 0.92 (0.84-1.01) | | 0.94 (0.85-1.04) |
| AHEI-2010 | | | | 0.93 (0.84-1.02) | | 0.93 (0.84-1.02) |
| HPDI | | | | 0.87 (0.79-0.95)* | | 0.89 (0.81-0.98)* |
| Breast cancer | 1906/64573 | 3.38 | | | | |
| AMED | | | | 0.96 (0.93-0.99)* | | 0.97 (0.94-1.003) |
| AEDII | | | | 0.99 (0.95-1.02) | | 0.99 (0.96-1.03) |
| AHEI-2010 | | | | 0.95 (0.92-0.98)* | | 0.95 (0.92-0.99)* |
| HPDI | | | | 0.97 (0.94-1.001) | | 0.98 (0.95-1.01) |
| Other cancers | 5189/108868 | 5.47 | | | | |
| AMED | | | | 0.91 (0.89-0.92)* | | 0.92 (0.91-0.94)* |
| AEDII | | | | 0.99 (0.97-1.004) | | 0.99 (0.97-1.01) |
| AHEI-2010 | | | | 0.93 (0.91-0.95)* | | 0.94 (0.92-0.96)* |
| HPDI | | | | 0.93 (0.91-0.95)* | | 0.95 (0.93-0.96)* |

between the MED diet and the risk of osteoporosis[24], more studies are needed to investigate the effects of other dietary scores. Limited data are available regarding the association between dietary scores and other chronic diseases. Our study also found an inverse association between some healthy dietary scores and the incidence of thyroid disorders, eczema, prostate disorders, cataract, and pernicious anemia, which provides further evidence on dietary guidelines for the prevention of these disorders.

Our study stands as the pioneering large cohort study to explore associations between four dietary scores and the risk of a wide range of chronic diseases. This study has several limitations. Firstly, while the web-based 24-hour dietary assessment tool

**Fig. 2 | The association between dietary scores and the risk of all cancers and types of cancers.** AEDII, Anti-Empirical Dietary Inflammatory Index; AHEI-2010, Alternate Healthy Eating Index-2010; AMED, Alternate Mediterranean Diet score; CI, confidence interval; HPDI, Healthful Plant-based Diet Index. The incidence refers to the number of event cases per 1000 person-years. All cancers encompass any type of cancer except for non-melanoma skin cancer. Cox proportional hazard regression models were used to examine associations of each of the four dietary scores with the risk of individual cancers. Model 1 was adjusted for age, sex, and total energy intake; Model 2 was adjusted for Model 1 plus ethnicity, education, income, BMI, smoking, sleep, physical activity, and GRS for longevity (pack-years, age stopping smoking, and number of cigarettes currently smoked daily were further adjusted for lung cancer). Dietary scores were analyzed as continuous variables (each quintile increment). The analysis for ovarian cancer and breast cancer was conducted among women only while the analysis for prostate cancer was conducted among men only. The vertical dash lines represent the hazard ratio of 1. Squares represent the hazard ratios (black color for AMED, orange color for AEDII, blue color for AHEI-2010, and green color for HPDI). Horizontal lines indicate the range of the 95% confidence interval. *Indicates a significant association through two-sided statistical tests while controlling for FDR.

employed in the UK Biobank study was validated against biomarkers, it is important to acknowledge the potential for measurement errors due to the self-reported nature. However, these measurement errors of diet are more likely to attenuate the true associations. Secondly, causal relationships cannot be established based on our results because of the observational nature of the study. Thirdly, incident cases of chronic diseases were identified using inpatient and mortality data, which might underestimate the incidence of these diseases. However, this is more likely to bias the associations towards the null. Fourthly, there may be detection bias for some diseases in the UK Biobank. For example, populations may vary in their likelihood of cancer detection due to differences in screening frequency, whilst cataracts may exhibit varying degrees of severity, but the available inpatient data in the UK might have limitations in accurately distinguishing these degrees. Even though our sensitivity analysis by excluding individuals who developed dementia within the initial four years of follow-up yielded results consistent with the main findings, it is worth considering that dementia could have begun prior to the diet assessment, given that the prodromal phase of dementia can extend over one decade[48]. Evidence suggests risk factors are different for clinically aggressive prostate cancer than for non-aggressive disease[49]. The inpatient and mortality data available in the UK Biobank do not differentiate between aggressive and non-aggressive prostate cancers, potentially introducing a bias into the relationship between dietary patterns and incident prostate cancer. Fifthly, we adjusted for the same confounders including demographic and lifestyle factors, BMI, energy intake, and genetic risk score (GRS) for longevity across all health conditions (besides lung cancer), which may be broader for some diseases. Sixthly, we cannot rule out the potential reverse causation between diet and psychological diseases as people in a situation or personalities prone to stress/anxiety could potentially adopt unhealthy dietary patterns[50] and thus were more likely to be diagnosed with psychological conditions during follow-up. Seventhly, investigating a broad range of chronic diseases offers certain benefits, but is also limited by narrowing the focus to a specific disease (discussion of the mechanisms). Finally, most of the participants in our study were Caucasians thus our findings may not be generalized to other ethnic groups.

In conclusion, greater adherence to healthy dietary patterns especially AMED is associated with a lower risk of multiple individual chronic diseases including all CMDs, some cancers, most psychological/neurological disorders, most digestive disorders, respiratory diseases, CKD, osteoporosis, eczema, prostate disorders, cataract, and pernicious anemia. Our findings suggest healthy dietary patterns may help prevent or delay the development of chronic diseases.

## Methods
### Study population
The UK Biobank is a population-based cohort of more than half million participants aged 39–70 years at enrollment (2006–2010)[51]. These participants were recruited from one of the 22 assessment centers throughout the United Kingdom. Details of the study design have been shown elsewhere[51].

The UK Biobank Study's ethical approval has been granted by the National Information Governance Board for Health and Social Care and the NHS North West Multicenter Research Ethics Committee (REC reference: 16/NW/0274). All participants provided informed consent through electronic signature at recruitment.

### Ascertainment of diseases
Diseases at baseline were defined if participants reported that they had ever been told by a doctor that they had the disease (Field code: Table S13). Additional disease cases at baseline were identified through inpatient records (Table S14). Forty-eight major diseases including CVDs (coronary heart disease, heart failure, atrial fibrillation, other cardiac disease, stroke, and peripheral vascular disease), diabetes, cancer (including melanoma, lung cancer, stomach cancer, oesophageal cancer, colon cancer, ovarian cancer, breast cancer, prostate cancer, and other cancers), COPD, neurodegenerative diseases, digestive disorders, and CKD were included in the analysis.

Incident cases of individual diseases were identified using inpatient hospital records and mortality registers. The inpatient hospital data were available since 1997 in the UK Biobank[51]. The international classification disease codes for each of the 48 diseases are listed in Table S14. The date of disease onset was defined as the earliest recorded date available. For each disease, individuals with the corresponding disease at baseline were excluded from the analysis. Person-years were computed by measuring the time from the baseline assessment date to the date of disease onset, date of death, or the conclusion of the follow-up period (December 31, 2020 for England and Wales and January 31, 2021 for Scotland), whichever came first.

### Dietary assessment
Diet was assessed using a web-based 24-h dietary assessment tool (the validated Oxford WebQ[52]. A sub-cohort of the UK Biobank completed the assessment on ≥1 of the five occasions between April 2009 and June 2012. Individuals who completed ≥2 dietary assessments were included in the analysis. The baseline assessment was established using the date of the dietary assessment from the most recent occasion.

We computed the amount of each food consumed by multiplying the assigned portion size by the quantity consumed. The nutrient amounts were computed by multiplying the quantity of each food consumed by the nutrient content of the portion (McCance and Widdowson's The Composition of Foods and its Supplements) and then summing this across all food groups[52]. The average food/nutrient intakes of the two or more dietary assessments was used in the analysis.

### Alternate Mediterranean diet score
The AMED developed by Fung et al. was calculated based on 9 food/nutrient groups (Table S15). For whole grains, vegetables (excluding potatoes), fruits, nuts and seeds, legumes, fish, and the ratio of

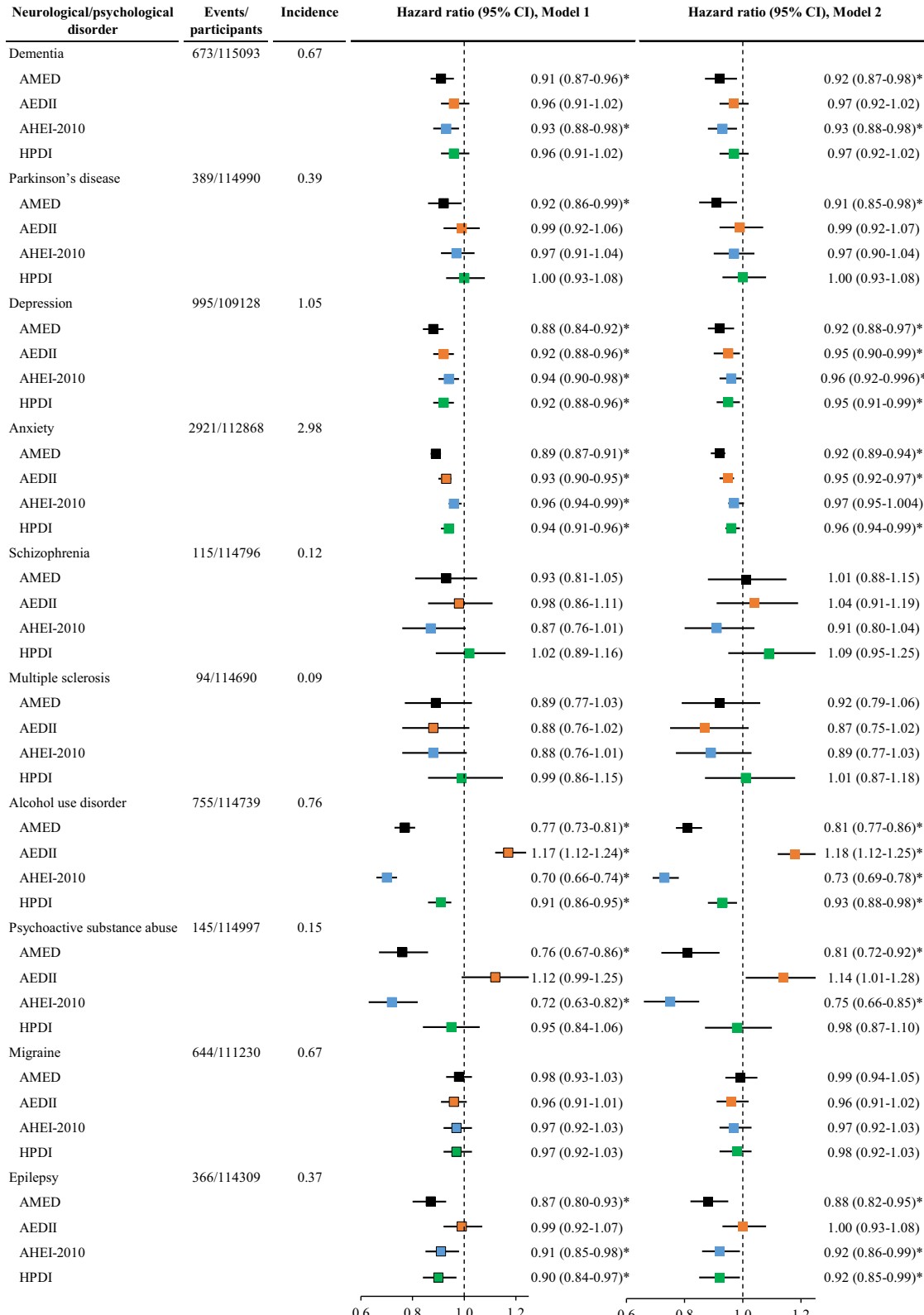

**Fig. 3 | The association between dietary scores and the risk of neurological and psychological disorders.** AEDII, Anti-Empirical Dietary Inflammatory Index; AHEI-2010, Alternate Healthy Eating Index-2010; AMED, Alternate Mediterranean Diet score; CI, confidence interval; HPDI, Healthful Plant-based Diet Index. The incidence refers to the number of event cases per 1000 person-years. Cox proportional hazard regression models were used to examine associations of each of the four dietary scores with the risk of individual neurological/psychological disorders. Model 1 was adjusted for age, sex, and total energy intake; Model 2 was adjusted for Model 1 plus ethnicity, education, income, BMI, smoking, sleep, physical activity, and GRS for longevity. Dietary scores were analyzed as continuous variables (each quintile increment). The vertical dash lines represent the hazard ratio of 1. Squares represent the hazard ratios (black color for AMED, orange color for AEDII, blue color for AHEI-2010, and green color for HPDI). Horizontal lines indicate the range of the 95% confidence interval. *Indicates a significant association through two-sided statistical tests while controlling for FDR.

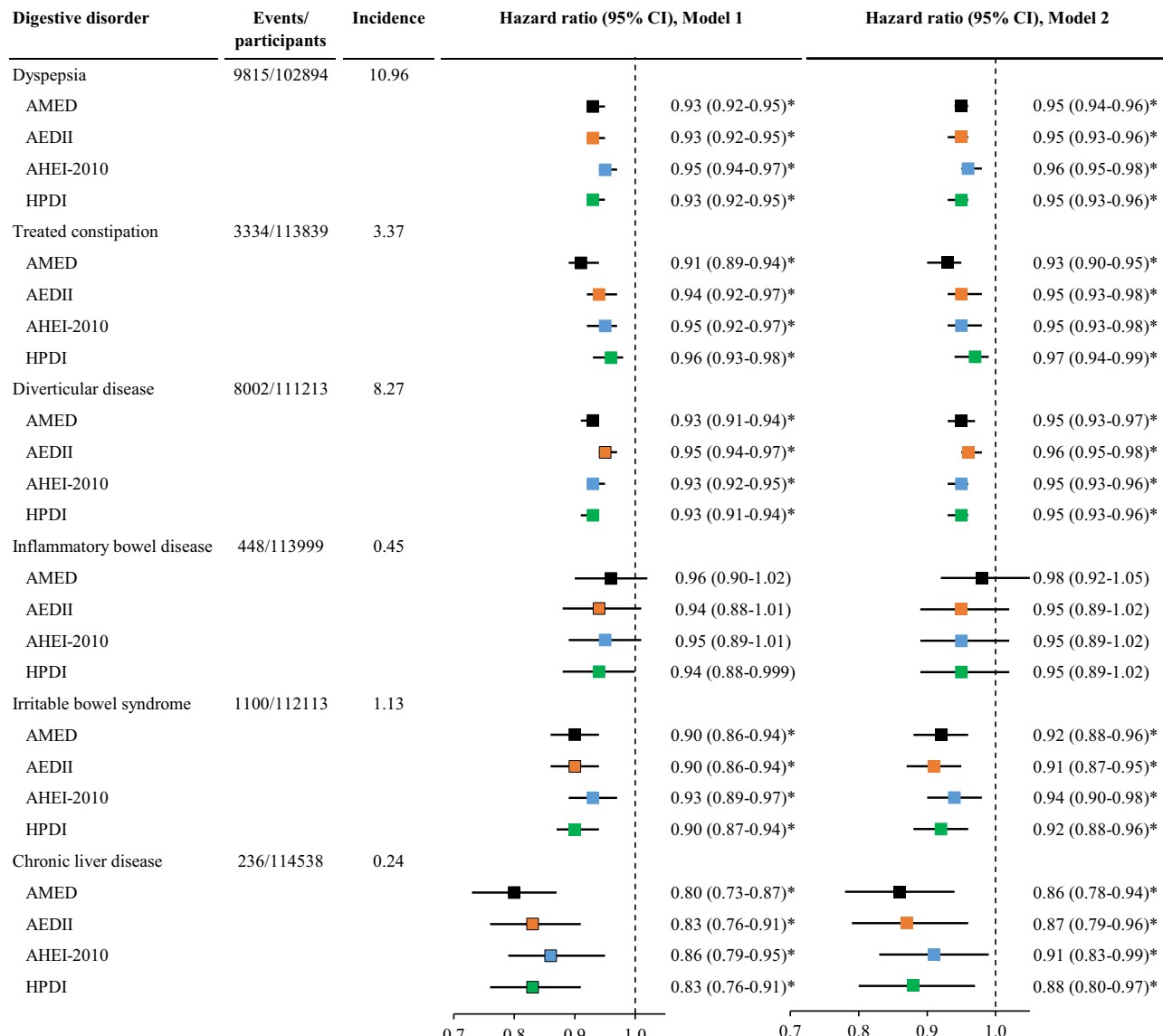

**Fig. 4 | The association between dietary scores and the risk of digestive disorders.** AEDII, Anti-Empirical Dietary Inflammatory Index; AHEI-2010, Alternate Healthy Eating Index-2010; AMED, Alternate Mediterranean Diet score; CI, confidence interval; HPDI, Healthful Plant-based Diet Index. The incidence refers to the number of event cases per 1000 person-years. Cox proportional hazard regression models were used to examine associations of each of the four dietary scores with the risk of individual digestive disorders. Model 1 was adjusted for age, sex, and total energy intake; Model 2 was adjusted for Model 1 plus ethnicity, education, income, BMI, smoking, sleep, physical activity, and GRS for longevity. Dietary scores were analyzed as continuous variables (each quintile increment). The vertical dash lines represent the hazard ratio of 1. Squares represent the hazard ratios (black color for AMED, orange color for AEDII, blue color for AHEI-2010, and green color for HPDI). Horizontal lines indicate the range of the 95% confidence interval. *Indicates a significant association through two-sided statistical tests while controlling for FDR.

monounsaturated fatty acid to saturated fatty acid, intakes above the sex-specific median of included participants were given 1 point and all other intakes were given 0 point. For meat (red and processed meats), 0 point was assigned to individuals above the median intake and 1 point to all others. Alcohol intake between 5 and 15 g/d was assigned 1 point. The total AMED score ranged from 0 to 9 with a higher score representing a healthier diet[53].

**Empirical dietary inflammatory index**

The EDII was calculated based on 18 food groups[54]. The score for each food was given according to their associations with interleukin-6, C-reactive protein, TNF-α receptor 2, and adiponectin[54]. To avert this scenario, the EDII score could become heavily reliant on just one or several components if individuals consumed too much of these food components, the maximum and minimum scores for individual food items were set at the levels given by previous dietary scores. Scores for the intakes between the maximum and minimum scores were proportionally calculated. The total EDII score was computed by summing the scores for the 18 food groups (Table S16). A higher EDII score indicates a higher pro-inflammatory diet. To make it comparable to other diet scores (higher score representing a healthier diet), an AEDII was created by reversing the EDII. A higher AEDII score indicates an anti-inflammatory diet.

**Alternate healthy eating index-2010**

The AHEI-2010 was calculated based on 10 food/nutrient groups in our analysis (whole grains, vegetables, fruits, nuts and legumes, sugar-sweetened beverages, and fruit juice, red/processed meat, fish (substitution of long-chain (n-3) fats), polyunsaturated fatty acid, adding

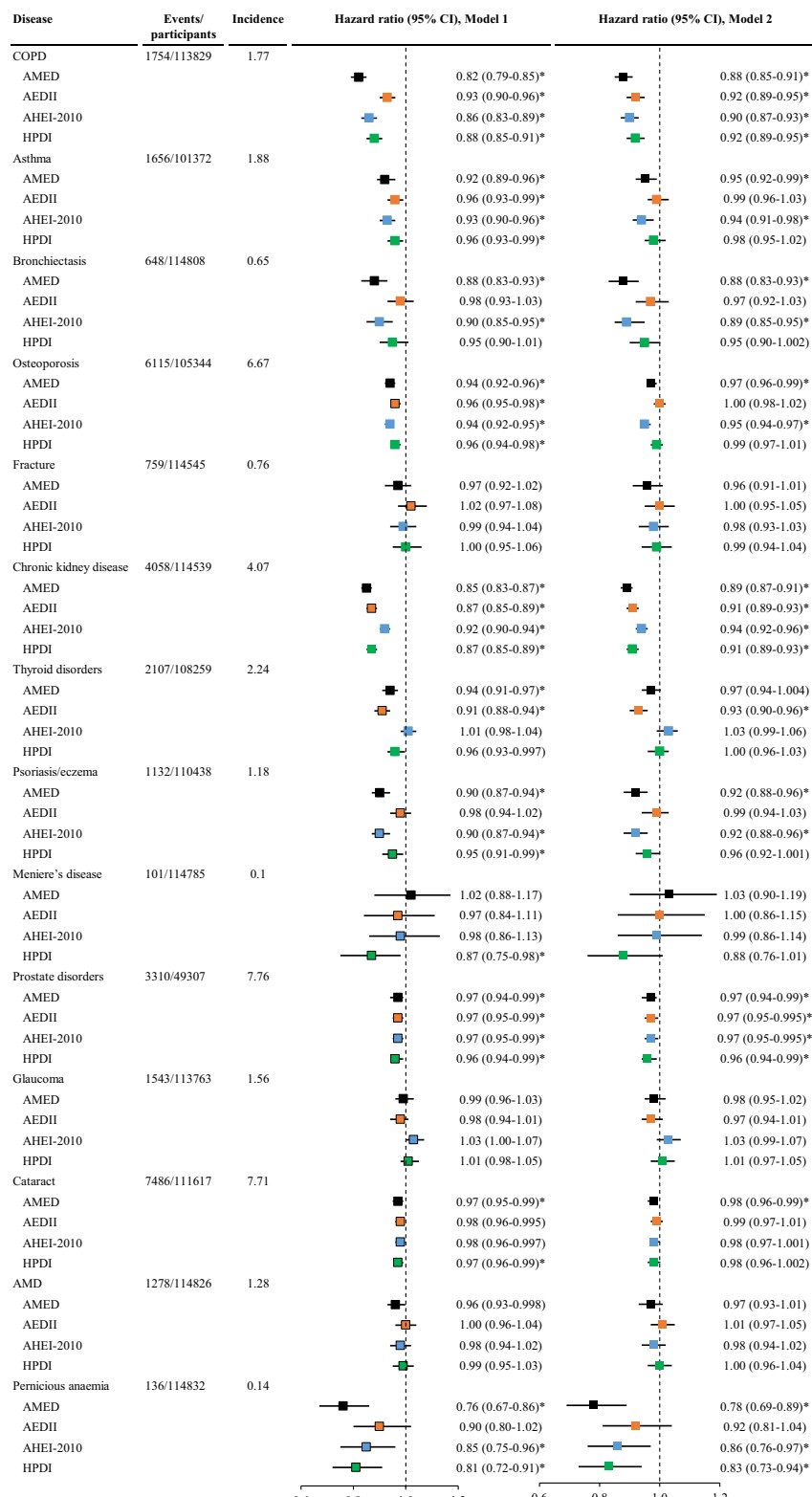

**Fig. 5 | The association between dietary scores and the risk of other chronic diseases.** AEDII, Anti-Empirical Dietary Inflammatory Index; AHEI-2010, Alternate Healthy Eating Index-2010; AMED, Alternate Mediterranean Diet score; AMD, age related macular degeneration; CI, confidence interval; COPD, chronic obstructive pulmonary disease; HPDI, Healthful Plant-based Diet Index. The incidence refers to the number of event cases per 1000 person-years. Cox proportional hazard regression models were used to examine associations of each of the four dietary scores with the risk of individual other chronic diseases. Model 1 was adjusted for age, sex, and total energy intake; Model 2 was adjusted for Model 1 plus ethnicity, education, income, BMI, smoking, sleep, physical activity, and GRS for longevity. Dietary scores were analyzed as continuous variables (each quintile increment). The analysis for prostate disorders was conducted among men only. The vertical dash lines represent the hazard ratio of 1. Squares represent the hazard ratios (black color for AMED, orange color for AEDII, blue color for AHEI-2010, and green color for HPDI). Horizontal lines indicate the range of the 95% confidence interval. *Indicates a significant association through two-sided statistical tests while controlling for FDR.

| Disease | Whole grains | Vegetable | Fruits | Legumes | Nuts and seeds | Fish | Red meat | Alcohol | MUFA to SFA ratio |
|---|---|---|---|---|---|---|---|---|---|
| **Cardiovascular disease** | 0.90 (0.86-0.94) | 0.90 (0.86-0.94) | 0.87 (0.83-0.91) | 0.89 (0.86-0.93) | 0.91 (0.87-0.95) | 0.88 (0.84-0.92) | 0.97 (0.93-1.02) | 0.93 (0.88-0.98) | 0.96 (0.92-1.00) |
| Coronary heart disease | 0.90 (0.85-0.96) | 0.92 (0.87-0.97) | 0.86 (0.81-0.91) | 0.91 (0.85-0.96) | 0.92 (0.87-0.98) | 0.89 (0.84-0.94) | 0.94 (0.89-1.00) | 0.94 (0.88-1.01) | 0.98 (0.92-1.03) |
| Heart failure | 0.80 (0.72-0.88) | 0.83 (0.75-0.92) | 0.84 (0.76-0.93) | 0.89 (0.80-0.99) | 0.80 (0.71-0.90) | 0.90 (0.81-0.99) | 0.96 (0.86-1.07) | 0.79 (0.69-0.91) | 0.93 (0.84-1.03) |
| Atrial fibrillation | 0.88 (0.80-0.97) | 0.91 (0.83-1.00) | 0.84 (0.76-0.92) | 0.86 (0.78-0.94) | 0.88 (0.80-0.98) | 0.85 (0.77-0.93) | 1.06 (0.96-1.17) | 0.89 (0.79-1.00) | 1.00 (0.92-1.10) |
| Other cardiac problem | 0.91 (0.85-0.98) | 0.87 (0.82-0.93) | 0.83 (0.78-0.89) | 0.89 (0.83-0.95) | 0.86 (0.80-0.92) | 0.86 (0.81-0.92) | 1.06 (0.98-1.13) | 0.95 (0.87-1.03) | 0.94 (0.88-1.00) |
| Stroke | 0.95 (0.81-1.10) | 0.75 (0.64-0.87) | 0.72 (0.62-0.83) | 0.79 (0.69-0.92) | 0.90 (0.76-1.06) | 0.82 (0.71-0.95) | 1.08 (0.92-1.26) | 0.86 (0.71-1.03) | 1.00 (0.87-1.16) |
| Peripheral vascular disease | 0.87 (0.76-1.00) | 0.89 (0.78-1.02) | 0.88 (0.77-1.01) | 0.83 (0.73-0.95) | 1.02 (0.88-1.17) | 0.80 (0.70-0.92) | 0.93 (0.81-1.08) | 0.97 (0.82-1.14) | 0.99 (0.86-1.13) |
| Hypertension | 0.85 (0.80-0.91) | 0.87 (0.82-0.93) | 0.85 (0.80-0.91) | 0.90 (0.84-0.96) | 0.88 (0.82-0.94) | 0.88 (0.83-0.94) | 0.93 (0.87-0.99) | 0.84 (0.78-0.91) | 1.01 (0.95-1.07) |
| Diabetes | 0.93 (0.85-1.01) | 0.84 (0.77-0.91) | 0.84 (0.77-0.91) | 0.92 (0.85-1.00) | 0.83 (0.76-0.91) | 0.85 (0.78-0.92) | 0.81 (0.74-0.89) | 0.95 (0.86-1.05) | 0.94 (0.87-1.02) |
| **All cancers** | 0.95 (0.90-1.00) | 0.84 (0.79-0.88) | 0.81 (0.77-0.85) | 0.90 (0.85-0.95) | 0.96 (0.91-1.01) | 0.91 (0.87-0.96) | 0.97 (0.92-1.03) | 0.91 (0.86-0.97) | 0.95 (0.91-1.00) |
| Non-melanoma skin cancer | 0.97 (0.90-1.03) | 0.91 (0.85-0.97) | 0.99 (0.93-1.06) | 0.97 (0.91-1.04) | 1.06 (0.99-1.14) | 0.97 (0.91-1.04) | 1.05 (0.98-1.12) | 0.97 (0.90-1.05) | 0.99 (0.93-1.06) |
| Melanoma | 1.04 (0.86-1.25) | 0.86 (0.71-1.03) | 0.82 (0.68-0.98) | 0.93 (0.78-1.11) | 1.11 (0.92-1.35) | 1.19 (0.99-1.43) | 1.08 (0.90-1.30) | 0.95 (0.76-1.18) | 1.07 (0.89-1.28) |
| Lung cancer | 0.80 (0.68-0.94) | 0.69 (0.59-0.81) | 0.69 (0.59-0.82) | 0.74 (0.63-0.87) | 0.90 (0.75-1.07) | 0.87 (0.74-1.02) | 1.02 (0.86-1.21) | 1.00 (0.82-1.22) | 0.84 (0.71-0.99) |
| Stomach cancer | 1.01 (0.72-1.42) | 0.69 (0.50-0.97) | 1.13 (0.81-1.58) | 1.19 (0.84-1.67) | 0.76 (0.52-1.10) | 0.98 (0.70-1.36) | 0.85 (0.59-1.22) | 0.76 (0.49-1.19) | 0.68 (0.48-0.94) |
| Oesophageal cancer | 0.87 (0.65-1.17) | 0.80 (0.60-1.06) | 0.87 (0.65-1.17) | 0.99 (0.74-1.31) | 0.99 (0.73-1.35) | 0.67 (0.50-0.89) | 0.82 (0.60-1.12) | 0.57 (0.38-0.86) | 0.73 (0.55-0.98) |
| Colon cancer | 0.94 (0.81-1.08) | 0.89 (0.77-1.03) | 0.90 (0.78-1.04) | 0.93 (0.81-1.07) | 0.89 (0.76-1.04) | 0.89 (0.77-1.02) | 0.98 (0.85-1.14) | 0.98 (0.83-1.17) | 1.01 (0.88-1.16) |
| Ovarian cancer | 0.94 (0.72-1.24) | 0.83 (0.63-1.08) | 0.77 (0.59-1.01) | 0.84 (0.65-1.10) | 0.85 (0.64-1.14) | 1.17 (0.89-1.55) | 1.13 (0.86-1.48) | 0.99 (0.72-1.36) | 0.86 (0.66-1.12) |
| Breast cancer | 1.02 (0.93-1.12) | 0.91 (0.83-1.00) | 0.86 (0.78-0.94) | 0.93 (0.85-1.02) | 1.00 (0.90-1.10) | 0.89 (0.82-0.98) | 1.03 (0.94-1.13) | 0.96 (0.86-1.07) | 0.98 (0.90-1.08) |
| Prostate cancer | 1.02 (0.93-1.12) | 0.95 (0.87-1.04) | 1.02 (0.93-1.11) | 0.95 (0.87-1.03) | 1.01 (0.92-1.11) | 1.03 (0.94-1.12) | 1.02 (0.93-1.12) | 1.05 (0.94-1.17) | 1.04 (0.96-1.14) |
| Other cancers | 0.96 (0.90-1.01) | 0.81 (0.77-0.86) | 0.81 (0.76-0.85) | 0.88 (0.83-0.93) | 0.95 (0.90-1.01) | 0.90 (0.85-0.95) | 0.98 (0.93-1.04) | 0.91 (0.85-0.98) | 0.95 (0.90-1.01) |
| Depression | 0.86 (0.75-0.97) | 0.84 (0.74-0.95) | 0.82 (0.72-0.93) | 0.86 (0.75-0.97) | 0.88 (0.76-1.01) | 0.79 (0.70-0.90) | 1.07 (0.94-1.22) | 0.90 (0.77-1.05) | 1.05 (0.92-1.19) |
| Anxiety | 0.91 (0.85-0.98) | 0.80 (0.74-0.86) | 0.79 (0.73-0.85) | 0.88 (0.81-0.94) | 0.97 (0.89-1.05) | 0.83 (0.77-0.89) | 1.08 (1.00-1.16) | 0.90 (0.82-0.98) | 0.99 (0.92-1.06) |
| Schizophrenia | 1.03 (0.70-1.50) | 0.87 (0.60-1.27) | 0.72 (0.49-1.06) | 0.69 (0.48-1.00) | 1.20 (0.82-1.77) | 1.12 (0.77-1.62) | 1.04 (0.71-1.53) | 0.80 (0.49-1.31) | 1.07 (0.74-1.54) |
| Alcohol use disorder | 0.62 (0.54-0.72) | 0.75 (0.65-0.87) | 0.58 (0.50-0.68) | 0.77 (0.67-0.89) | 0.96 (0.82-1.12) | 0.88 (0.77-1.02) | 0.83 (0.71-0.98) | 0.31 (0.24-0.41) | 1.11 (0.96-1.28) |
| Psychoactive substance abuse | 0.51 (0.37-0.70) | 0.81 (0.58-1.13) | 0.71 (0.51-0.98) | 0.70 (0.51-0.95) | 0.79 (0.56-1.13) | 0.72 (0.52-1.00) | 1.18 (0.84-1.66) | 0.47 (0.27-0.80) | 0.99 (0.73-1.35) |
| Epilepsy | 0.79 (0.64-0.97) | 0.79 (0.64-0.98) | 0.61 (0.49-0.75) | 0.79 (0.65-0.98) | 0.87 (0.69-1.09) | 0.80 (0.65-0.99) | 1.07 (0.87-1.33) | 0.96 (0.74-1.23) | 1.04 (0.85-1.28) |
| Migraine | 0.95 (0.81-1.12) | 0.79 (0.67-0.92) | 0.84 (0.72-0.99) | 1.07 (0.92-1.26) | 1.12 (0.94-1.32) | 0.92 (0.78-1.07) | 1.08 (0.92-1.27) | 1.04 (0.86-1.25) | 1.05 (0.90-1.23) |
| Dementia | 0.81 (0.69-0.94) | 0.87 (0.75-1.02) | 0.94 (0.80-1.10) | 0.88 (0.76-1.03) | 1.00 (0.85-1.18) | 0.88 (0.76-1.03) | 1.06 (0.91-1.24) | 0.98 (0.81-1.18) | 0.91 (0.78-1.06) |
| Parkinson's disease | 0.96 (0.78-1.18) | 0.76 (0.62-0.93) | 1.09 (0.89-1.34) | 0.83 (0.68-1.01) | 0.75 (0.59-0.94) | 1.07 (0.87-1.31) | 0.97 (0.79-1.20) | 0.94 (0.74-1.21) | 0.85 (0.69-1.03) |
| Multiple sclerosis | 1.09 (0.72-1.65) | 0.79 (0.52-1.19) | 1.30 (0.86-1.97) | 0.83 (0.55-1.25) | 0.93 (0.59-1.46) | 0.75 (0.50-1.12) | 1.17 (0.77-1.77) | 0.90 (0.53-1.52) | 0.66 (0.43-0.99) |
| Bronchiectasis | 0.87 (0.75-1.02) | 0.78 (0.67-0.92) | 0.81 (0.69-0.95) | 0.83 (0.71-0.98) | 0.81 (0.68-0.97) | 0.86 (0.73-1.00) | 1.20 (1.03-1.41) | 1.05 (0.87-1.27) | 0.80 (0.68-0.93) |
| Asthma | 0.97 (0.88-1.08) | 0.90 (0.82-0.99) | 0.87 (0.79-0.96) | 0.86 (0.78-0.94) | 0.84 (0.76-0.94) | 0.96 (0.87-1.06) | 1.04 (0.94-1.15) | 0.89 (0.79-1.01) | 0.88 (0.80-0.97) |
| COPD | 0.76 (0.69-0.83) | 0.87 (0.79-0.96) | 0.73 (0.67-0.81) | 0.94 (0.86-1.04) | 0.83 (0.74-0.92) | 0.78 (0.71-0.85) | 0.96 (0.87-1.07) | 0.89 (0.79-1.01) | 0.89 (0.81-0.98) |
| CKD | 0.85 (0.79-0.90) | 0.76 (0.71-0.81) | 0.76 (0.71-0.81) | 0.85 (0.80-0.91) | 0.88 (0.82-0.94) | 0.79 (0.75-0.85) | 0.91 (0.85-0.97) | 0.93 (0.87-1.01) | 0.92 (0.86-0.98) |
| Chronic liver disease | 0.75 (0.58-0.97) | 0.80 (0.62-1.04) | 0.69 (0.53-0.91) | 0.62 (0.48-0.80) | 0.73 (0.54-0.99) | 0.85 (0.66-1.10) | 0.79 (0.60-1.05) | 0.76 (0.54-1.08) | 1.01 (0.78-1.31) |
| Irritable bowel syndrome | 0.81 (0.72-0.92) | 0.85 (0.75-0.96) | 0.77 (0.68-0.87) | 0.82 (0.73-0.93) | 1.02 (0.90-1.17) | 0.89 (0.79-1.01) | 0.98 (0.87-1.11) | 1.04 (0.90-1.20) | 0.96 (0.85-1.08) |
| Inflammatory bowel disease | 0.88 (0.73-1.07) | 0.94 (0.78-1.14) | 0.92 (0.76-1.11) | 0.91 (0.75-1.10) | 1.05 (0.86-1.28) | 0.95 (0.79-1.15) | 1.12 (0.92-1.36) | 1.02 (0.81-1.28) | 0.99 (0.82-1.19) |
| Treated constipation | 0.91 (0.85-0.97) | 0.83 (0.78-0.89) | 0.84 (0.78-0.90) | 0.86 (0.80-0.92) | 0.96 (0.89-1.03) | 0.84 (0.79-0.90) | 1.07 (0.99-1.14) | 0.98 (0.90-1.07) | 0.92 (0.86-0.98) |
| Dyspepsia | 0.92 (0.88-0.96) | 0.89 (0.86-0.93) | 0.84 (0.80-0.87) | 0.95 (0.91-0.99) | 0.96 (0.92-1.00) | 0.90 (0.86-0.93) | 1.00 (0.96-1.04) | 0.96 (0.91-1.00) | 0.98 (0.94-1.01) |
| Diverticular disease | 0.89 (0.85-0.93) | 0.93 (0.89-0.97) | 0.84 (0.80-0.87) | 0.96 (0.92-1.00) | 0.93 (0.88-0.97) | 0.96 (0.92-1.01) | 0.92 (0.87-0.96) | 0.92 (0.88-0.98) | 0.95 (0.91-0.99) |
| Pernicious anaemia | 0.70 (0.50-0.99) | 0.70 (0.50-0.99) | 0.69 (0.49-0.98) | 0.69 (0.49-0.96) | 0.62 (0.41-0.94) | 0.76 (0.54-1.07) | 0.97 (0.68-1.38) | 1.06 (0.70-1.62) | 0.86 (0.61-1.21) |
| Fracture | 1.01 (0.87-1.17) | 0.96 (0.83-1.12) | 1.07 (0.92-1.23) | 0.90 (0.78-1.04) | 0.99 (0.85-1.15) | 0.99 (0.85-1.14) | 1.09 (0.94-1.27) | 0.90 (0.75-1.07) | 0.91 (0.79-1.05) |
| Osteoporosis | 0.84 (0.80-0.89) | 0.97 (0.92-1.02) | 0.97 (0.92-1.03) | 0.95 (0.90-1.00) | 0.92 (0.87-0.97) | 0.91 (0.86-0.96) | 0.92 (0.87-0.97) | 0.91 (0.85-0.96) | 0.93 (0.88-0.98) |
| Meniere's disease | 0.62 (0.42-0.93) | 0.72 (0.48-1.08) | 1.07 (0.72-1.59) | 0.75 (0.51-1.11) | 1.33 (0.88-2.01) | 1.07 (0.72-1.59) | 1.63 (1.10-2.43) | 1.11 (0.70-1.77) | 1.23 (0.83-1.83) |
| Eczema | 0.83 (0.74-0.94) | 0.82 (0.72-0.92) | 0.79 (0.70-0.89) | 0.95 (0.84-1.07) | 0.92 (0.81-1.05) | 0.89 (0.79-1.00) | 1.06 (0.94-1.20) | 0.97 (0.84-1.12) | 0.95 (0.85-1.07) |
| Glaucoma | 1.00 (0.90-1.11) | 1.05 (0.95-1.17) | 0.99 (0.89-1.10) | 0.99 (0.89-1.10) | 1.05 (0.94-1.17) | 1.01 (0.91-1.12) | 0.96 (0.86-1.06) | 0.92 (0.82-1.05) | 0.98 (0.88-1.08) |
| Cataract | 0.97 (0.93-1.02) | 0.98 (0.93-1.03) | 0.99 (0.94-1.04) | 0.96 (0.92-1.01) | 1.02 (0.97-1.08) | 0.93 (0.88-0.97) | 0.94 (0.90-0.99) | 0.97 (0.92-1.02) | 0.99 (0.95-1.04) |
| AMD | 0.93 (0.83-1.05) | 0.99 (0.89-1.11) | 0.94 (0.84-1.06) | 0.97 (0.87-1.09) | 1.10 (0.97-1.23) | 0.96 (0.86-1.08) | 0.96 (0.86-1.08) | 0.96 (0.84-1.10) | 1.02 (0.91-1.14) |
| Thyroid disorders | 0.96 (0.88-1.05) | 0.98 (0.89-1.07) | 0.84 (0.77-0.92) | 0.94 (0.86-1.02) | 0.95 (0.86-1.04) | 0.89 (0.82-0.97) | 1.03 (0.94-1.12) | 1.00 (0.91-1.11) | 0.92 (0.84-1.00) |
| Prostate disorders | 0.96 (0.89-1.03) | 0.89 (0.83-0.95) | 0.94 (0.88-1.01) | 0.94 (0.88-1.01) | 0.95 (0.88-1.03) | 0.94 (0.87-1.00) | 1.06 (0.99-1.14) | 1.07 (0.98-1.17) | 0.99 (0.92-1.06) |

salt to foods (substitution of sodium), and alcohol)[55]. Scores for the intakes between 0 and 10 were proportionately calculated. Each component was scored from 0 to 10 and the total AHEI-2010 score ranged from 0 to 100 (Table S17). Trans fatty acid was not included in the calculation as it was not available in the study.

## Healthful plant-based diet index

The HPDI was computed based on 17 food groups[56,57]. Vegetable oil was not included in the calculation given it was not available in the UK Biobank[57]. A score between 1 and 5 was assigned to the quintiles of each of the 17 food groups. For plant-based food groups (whole grains,

**Fig. 6 | The association between individual components of the Alternate Mediterranean Diet Index and the risk of individual chronic diseases.** AMD, age related macular degeneration; CKD, chronic kidney disease; COPD, chronic obstructive pulmonary disease. Data are hazard ratios (95% confidence intervals). All cancers encompass any type of cancer except for non-melanoma skin cancer. Cardiovascular disease includes coronary heart disease, heart failure, atrial fibrillation, other cardiac disease, stroke, and peripheral vascular disease. All cancers encompass any type of cancer except for non-melanoma skin cancer. Cox proportional hazard regression models were used to examine associations of individual components of the Alternate Mediterranean Diet Index with the risk of individual chronic diseases. The analysis was adjusted for age, sex, ethnicity, education, income, smoking, alcohol consumption, sleep, physical activity, GRS for longevity, and total energy intake. The hazard ratio refers to the risk for the disease associated with the component (recommended level versus non-recommended level). The analysis for ovarian cancer and breast cancer was conducted among women only while the analysis for prostate cancer and prostate disorders was conducted among men only. Green color refers to inverse associations and red color refers to positive associations. We conducted two-sided statistical tests, and significant associations were adjusted for FDR.

fruits, vegetables, nuts, legumes, tea and/or coffee) a score of 5 was assigned to the highest quintile and 1 to the lowest quintile. For animal-based food groups (animal fat, dairy, eggs, fish/seafood, meat, miscellaneous animal-based foods), a score of 1 was assigned to the highest quintile and 5 to the lowest quintile. For other food groups (refined grains, potatoes, sugary drinks, fruit juices, sweets and/or desserts), a score of 1 was assigned to the highest quintile (Table S18). The total HPDI score ranged from 17 to 85 with a higher score representing a healthier diet.

## Covariates

We obtained information on age, sex, ethnicity, education, household income, alcohol consumption, physical activity, smoking, sleep duration, and medical history through questionnaires on a touch-screen computer. Sleep duration was assessed based on the question "About how many hours' sleep do you get in every 24 h?" Physical activity was assessed using a short form of the International Physical Activity Questionnaire. GRS for longevity was computed using 78 single-nucleotide polymorphisms with a higher score representing longer potential longevity[58]. We adjusted for GRS in the analysis given that it predicted an individual's genetic predisposition to experiencing a lower risk of age-related diseases.

## Statistical analysis

Baseline characteristics were expressed as frequency (percentage) or means ± (SDs) by quintiles of dietary scores. We used ANOVA for continuous variables and Chi-square test for categorical variables to test the difference of characteristics across quintiles of dietary scores.

As most of the associations between dietary patterns and the risk of individual chronic diseases did not differ significantly between sexes, we reported the results for the whole population. Cox proportional hazard regression models were used to examine associations of each of the four dietary scores with the incidence of each of the 48 chronic diseases adjusted for potential confounding variables. We tested three models: (1) Model 1 was adjusted for age, sex, and total energy intake; (2) Model 2 was adjusted for Model 1 plus ethnicity, education, income, BMI, smoking, sleep, physical activity, and GRS for longevity (pack-years, age stopping smoking, and number of cigarettes currently smoked daily were further adjusted for lung cancer). Dietary scores were analyzed as both categorical variables (quintiles) and continuous variables (each quintile increment). The association between components for the dietary score that was predictive of more chronic diseases and the incidence of chronic diseases was examined using Cox proportional hazard regression models.

Whether the association between dietary scores and the risk of individual diseases was moderated by important confounders including age, sex, education, obesity, hypertension, diabetes, dyslipidemia, GRS for longevity, and education was then analyzed.

A sensitivity analysis of the association between dietary scores and the risk of individual diseases was conducted among individuals by excluding individuals who developed the corresponding disease in the first 4 years of follow-up. A further sensitivity analysis was conducted among individuals who completed ≥3 dietary assessments.

The percentages of participants with missing values on physical activity, education, income, smoking, and BMI, were 14.1%, 0.3%, 11.1%, 0.2%, and 1.4%, respectively. Multiple imputations for missing data were conducted to create 10 imputed datasets.

Data analyses were performed using SAS 9.4 for Windows (SAS Institute Inc.), and all *P* values were two-sided, with a significance level set at <0.05. In our study, we linked four distinct dietary patterns to the risk of 48 different chronic diseases. For these multiple comparisons, Benjamin-Hochberg's procedure was used to control the FDR at a 5% level[59].

## Reporting summary

Further information on research design is available in the Nature Portfolio Reporting Summary linked to this article.

## Data availability

All the analyses are conducted based on the UK Biobank data. The UK Biobank dataset used in this study is not publicly available but can be obtained by application through the data-access protocol (https://www.ukbiobank.ac.uk/). The typical duration from submitting an application to the release of data is approximately 15 weeks for the UK Biobank. The data used in this study is available in the UK Biobank database under the application number of 62443 and 62489.

## Code availability

The codes used for analyses in this study are available upon request. Access to codes will be granted for requests for academic use within 3 weeks of application by contacting Dr. Xianwen Shang (andy243@126.com; https://github.com/Xianwenshang2023/Diet-scores-and-chronic-diseases).

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

## Acknowledgements
This research was conducted using the UK Biobank resource. We thank the participants of the UK Biobank. X.l.Z. receives GDPH Supporting Fund for Talent Program (KJ2020633). Z.Z. receives support from the National Natural Science Foundation of China (82101173), the Research Foundation of Medical Science and Technology of Guangdong Province (B2021237). H.Y. receives support from the National Natural Science Foundation of China (81870663, 82171075), the Outstanding Young Talent Trainee Program of Guangdong Provincial People's Hospital (KJ012019087), Guangdong Provincial People's Hospital Scientific Research Funds for Leading Medical Talents and Distinguished Young Scholars in Guangdong Province (KJ012019457), Talent Introduction Fund of Guangdong Provincial People's Hospital (Y012018145). M.H. receives support from the High-level Talent Flexible Introduction Fund of Guangdong Provincial People's Hospital (No. KJ012019530). M.H. also receives support from the University of Melbourne at Research Accelerator Program and the CERA Foundation. The Center for Eye Research Australia receives Operational Infrastructure Support from the Victorian State Government. The sponsor or funding organization had no role in the design or conduct of this research. The sponsor or funding organization had no role in the design, conduct, analysis, or reporting of this study. The funding sources did not participate in the design and conduct of the study; collection, management, analysis, and interpretation of the data; preparation, review, or approval of the manuscript; and decision to submit the manuscript for publication.

## Author contributions
X.S., M.H. conceived and designed the study. Z.Z., W.W. performed data curation. X.S. conducted data analysis and drafted the initial manuscript. X.S., J.L., X.l.Z., Y.u.H., S.L., Z.Z., W.W., X.y.Z., S.T., Y.i.H., Z.G., H.Y., and M.H. made a critical revision to the manuscript for important intellectual content. All authors read the manuscript and approved the final draft. X.S. and M.H. are study guarantors. The corresponding authors attest that all listed authors meet authorship criteria and that no others meeting the criteria have been omitted.

## Competing interests
The authors declare no competing interests.
