## [Peer Review File · Nature Communications]

Healthy dietary patterns and the risk of individual chronic diseases in community-dwelling adultsREVIEWER COMMENTS

Reviewer #1 (Remarks to the Author):

This manuscript is well-written and generally interesting. It builds upon previous analyses on dietary patterns and extends to a comprehensive list of diseases. Even though only four dietary patterns were included, they are widely recommended and might be well-accepted by the public. The paper also provides more evidence for the protective role of healthy diet in preventing several under-studied diseases such as chronic kidney disease, endocrine disorders etc. However, I have some concerns that need to be addressed.

Major comments:

Given that all these diseases were included under the umbrella term of “chronic disease”, the authors need to clarify how they define “chronic diseases”. The introduction includes a nice discussion about healthy ageing. However, it is not clear to me how diseases such as dyspepsia, alcohol use disorder etc. might be related to this general topic. The title suggests an investigation of co-occurrence of two or more chronic diseases, which is different from the aim of present study. The conclusion highlights prevention of age-related chronic diseases although some diseases under investigation are not necessarily age-related, e.g., schizophrenia.

A strength and a limitation of the approach is the examination of a wide spectrum of diseases. The advantage is that this allows one to examine the effects of diet more broadly rather than focusing on an individual or a few diseases. The downside is that one cannot do the range of analyses to examine associations in more detail. For example, there could be detection bias for some diseases (e.g., cancers, based on screening, and diseases like cataracts based on diagnostic intensity); latencies/induction periods likely vary for many of the diseases; some diseases like dementia may require a long time lag to reliably see effects; while diet has many plausible links to diseases such as metabolic diseases, for others it is less clear; the list of confounders may be broader for some diseases. There are disease-specific nuances that cannot be examined (e.g., for prostate cancer, most evidence suggests that risk factors are different for clinically aggressive disease than for total disease). The broad view has advantages though what is sacrificed should be discussed because readers interested in a specific disease will focus on that disease.

In regards to validity, two important concerns are 1) measurement error of diet (which would tend to attenuate any true associations), and 2) residual confounding, which can impact findings in less predictable ways. Residual confounding could occur by missing some important confounders and by measurement error in the measured ones. For example, it appears that smoking was controlled as

never, past, current. There could be residual confounding, esp for the strongly smoking-related outcomes like lung cancer, based on pack-years, when quit, dose in current smokers. If possible, this can be looked at. An interesting analysis would be to examine never smokers, which would remove any potential confounding by smoking.

Personally, to reduce the amount of data to look at, I would prefer two models, base model with age, sex, and energy intake, and one with everything. I view energy intake not as a confounder per se, but as an important variable to adjust for to reduce measurement error in diet. Also, in part, energy intake is determined by body size and physical activity (e.g., a large, active male would have much higher energy requirement than a small, inactive female) so energy intake (esp in balance) is not a true confounder (independent variable in another causal pathway). At least some comment would be useful in what the different models show (e.g. if HRs don't change much, that might be support for less residual confounding, etc.) As such, 3 models are shown which triples the data presented but the reason why different models are done is not specified.

Did BMI influence the associations between dietary patterns and outcomes? The paper mentions BMI in the text but none of the tables/figures include BMI. BMI could be considered a confounder or a mediator.

I might be cautious between a causal connection between diet and psychologic diseases such as depression. There is a link between mental states and poor diets. For e.g., people in a situation/circumstances or personalities prone to stress/anxiety may have poor diets and then over time, some of these people would be more likely to be diagnosed (e.g., with depression).

Minor comments:

Line 137. The scoring method for EDII as shown in Table S4 is different from what it was originally developed. The original score is a weighted sum of 18 food groups (as continuous variables) multiplied by specific weights. It seems the authors have modified EDII. I think the authors need to emphasize this in the methods and explain why they modified it.

Line 50. Second sentence of the Intro is incomplete.

Line 52. How much is the total number of deaths increased due to increasing size of older population?

Line 184.

- Some dietary patterns include alcohol component. Inclusion of alcohol intake as a covariate might lead to overadjustment.

- The authors need to justify why they include GRS for longevity as a covariate for different diseases. In Table 1, GRS has little variation across dietary score quintiles (as would be expected).

Line 204. I would appreciate a transparent reporting of number of hypotheses being considered.

Line 230-235: These numerical values are getting crowded. If you have included them in figures, you don't need to mention all of them.

Line 299" I thought the Mediterranean diet supported higher fish intake, but here it is stated that lower fish intake contributed to the benefit?

Fig1-2. Some associations appeared marginally significant (95% CI includes 1). It's not clear why the authors reported them to be statistically significant even after FDR correction.

It is not clear how many cases were included for each disease. Does total cancer include non-melanoma skin cancer?

The text mentions about average follow-up years for two diseases. I think a supplementary table of median follow-up (and IQR) for each disease will be useful.

Fig 6. I wonder why the authors only look at components of alternate Mediterranean diet. I can see that AMED was inversely associated with the largest number of diseases in this paper. Did the authors also consider effect size and overall disease burden when interpreting the results?

Fig S1-4. I think all results from subgroup analyses should be presented.

Line 328. Reference 30 seems to be incorrect.

Line 342. While some items may overlap between AMED and ketogenic diet, AMED is not ketogenic. Thus, any advantage of the AMED diet is not related to ketogenesis.

Some findings related to the AEDII are likely due to the fact that alcohol is in the score (e.g. association with alcohol use disorder, psychoactive substance abuse; weak effect for lung cancer because alcohol drinkers tend to smoke too).

Reviewer #2 (Remarks to the Author):

The paper has well designed; analyses were appropriate. More evidence related healthy diet as a protective factor. However, these scores are based on different criteria, and it is unclear which score best predicts chronic disease risk. There is a lot of information which was well summarised, but we lost the importance of diet in each disease, including all in the same paper. Analyses were adjusted by confounding, but maybe moderator analysis by obesity, education, or other factors specific for each disease, which would be interesting too and it's lost.

Reviewer #1 (Remarks to the Author):

This manuscript is well-written and generally interesting. It builds upon previous analyses on dietary patterns and extends to a comprehensive list of diseases. Even though only four dietary patterns were included, they are widely recommended and might be well-accepted by the public. The paper also provides more evidence for the protective role of healthy diet in preventing several under-studied diseases such as chronic kidney disease, endocrine disorders etc. However, I have some concerns that need to be addressed.

Response:

We thank the Reviewer for his or her positive comments on our manuscript. We agree with the Reviewer that some concerns require attention and we have revised the manuscript accordingly.

Major comments:

Given that all these diseases were included under the umbrella term of “chronic disease”, the authors need to clarify how they define “chronic diseases”. The introduction includes a nice discussion about healthy ageing. However, it is not clear to me how diseases such as dyspepsia, alcohol use disorder etc. might be related to this general topic. The title suggests an investigation of co-occurrence of two or more chronic diseases, which is different from the aim of present study. The conclusion highlights prevention of age-related chronic diseases although some diseases under investigation are not necessarily age-related, e.g., schizophrenia.

Response:

We thank the Reviewer for raising this important concern. We agree with the Reviewer that non-age-related chronic diseases such as alcohol use disorder, other psychoactive substance abuse, schizophrenia, migraine, multiple sclerosis, dyspepsia, eczema, irritable bowel syndrome, and inflammatory bowel disease were included in our analyses.

We have added the information regarding these non-age-related diseases:

In addition to these age-related chronic diseases, psychiatric/neurological disorders including alcohol use disorder, other psychoactive substance abuse, schizophrenia, multiple sclerosis, and migraine have been linked to elevated mortality risks.⁵⁻⁹ While digestive disorders like dyspepsia, irritable bowel syndrome, and inflammatory bowel disease might not directly contribute to mortality risk, these conditions are widespread and place a significant burden on healthcare and economic systems.¹⁰⁻¹² Therefore, investigating significant modifiable factors for these non-age-related chronic conditions also holds considerable interest.

We agree with the Reviewer that the title was misleading, and we have revised the title to “Healthy dietary patterns and the risk of individual chronic diseases in community-dwelling adults”.

We have revised the Conclusion as:

Greater adherence to healthy dietary patterns especially AMED is associated with a lower risk of multiple individual chronic diseases including all CMDs, some cancers, most psychological/neurological disorders, most digestive disorders, respiratory diseases, chronic

kidney disease, osteoporosis, eczema, prostate disorders, cataract, and pernicious anaemia. Our findings support dietary guidelines for the prevention of chronic diseases.

A strength and a limitation of the approach is the examination of a wide spectrum of diseases. The advantage is that this allows one to examine the effects of diet more broadly rather than focusing on an individual or a few diseases. The downside is that one cannot do the range of analyses to examine associations in more detail. For example, there could be detection bias for some diseases (e.g., cancers, based on screening, and diseases like cataracts based on diagnostic intensity); latencies/induction periods likely vary for many of the diseases; some diseases like dementia may require a long time lag to reliably see effects; while diet has many plausible links to diseases such as metabolic diseases, for others it is less clear; the list of confounders may be broader for some diseases. There are disease-specific nuances that cannot be examined (e.g., for prostate cancer, most evidence suggests that risk factors are different for clinically aggressive disease than for total disease). The broad view has advantages though what is sacrificed should be discussed because readers interested in a specific disease will focus on that disease.

Response:

We agree with the Reviewer that there are both pros and cons to investigating a broad range of diseases. We have mentioned this as a limitation:

Sixthly, investigating a broad range of chronic diseases offers certain benefits, but is also limited by narrowing the focus to a specific disease (discussion of the mechanisms).

The Reviewer is correct that there may be detection bias for some diseases. We have added this as a limitation:

Fourthly, there may be detection bias for some diseases in the UK Biobank. For example, populations may vary in their likelihood of cancer detection due to differences in screening frequency, whilst cataracts may exhibit varying degrees of severity, but the available inpatient data in the UK might have limitations in accurately distinguishing these degrees. Even though our sensitivity analysis by excluding individuals who developed dementia within the initial four years of follow-up yielded results consistent with the main findings, it is worth considering that dementia could have begun prior to the diet assessment, given that the prodromal phase of dementia can extend over one decade.⁵⁵ Evidence suggests risk factors are different for clinically aggressive prostate cancer than for non-aggressive disease.⁵⁶ The inpatient and mortality data available in the UK Biobank do not differentiate between aggressive and non-aggressive prostate cancers, potentially introducing a bias into the relationship between dietary patterns and incident prostate cancer.

We also agree with the Reviewer that the list of confounders adjusted for all individual diseases may be broader for some diseases. We have discussed this:

Fifthly, we adjusted for the same confounders including demographic and lifestyle factors, BMI, energy intake, and GRS for longevity across all health conditions (besides lung cancer), which may be broader for some diseases.

In regards to validity, two important concerns are 1) measurement error of diet (which would tend to attenuate any true associations), and 2) residual confounding, which can impact findings in less

predictable ways. Residual confounding could occur by missing some important confounders and by measurement error in the measured ones. For example, it appears that smoking was controlled as never, past, current. There could be residual confounding, esp for the strongly smoking-related outcomes like lung cancer, based on pack-years, when quit, dose in current smokers. If possible, this can be looked at. An interesting analysis would be to examine never smokers, which would remove any potential confounding by smoking.

Response:

We agree with the Reviewer that there are measurement errors in diet and mentioned this as a limitation:

Firstly, while the web-based 24-hour dietary assessment tool employed in the UK Biobank study was validated against biomarkers, it is important to acknowledge the potential for measurement errors due to the self-reported nature. However, these measurement errors of diet are more likely to attenuate the true associations.

As the Reviewer suggests, we further adjusted for pack-years, the age stopping smoking, and the number of cigarettes currently smoked daily for lung cancer. In the full model, AMED and AHEI-2010 were inversely associated with the risk of lung cancer. In addition, we have conducted further analysis for the association between dietary patterns and incident lung cancer stratified by smoking (never smokers, current/former smokers). The hazard ratios (95% CI) for lung cancer associated with AMED among never smokers, former smokers, and current smokers were 0.94 (0.84-1.05), 0.81 (0.74-0.88), and 0.78 (0.70-0.88), respectively (raw P-value for interaction=0.0307). The corresponding numbers for AHEI-2010 were 0.93 (0.83-1.04), 0.89 (0.82-0.97), and 0.80 (0.71-0.90), respectively (raw P-value for

interaction=0.0806). Given the multiple comparisons, the interactions were not significant after controlling false discovery rate.

Personally, to reduce the amount of data to look at, I would prefer two models, base model with age, sex, and energy intake, and one with everything. I view energy intake not as a confounder per se, but as an important variable to adjust for to reduce measurement error in diet. Also, in part, energy intake is determined by body size and physical activity (e.g., a large, active male would have much higher energy requirement than a small, inactive female) so energy intake (esp in balance) is not a true confounder (independent variable in another causal pathway). At least some comment would be useful in what the different models show (e.g. if HRs don't change much, that might be support for less residual confounding, etc.) As such, 3 models are shown which triples the data presented but the reason why different models are done is not specified.

Response:

We thank the Reviewer for raising this important concern.

We agree with the Reviewer that energy intake is determined by body size and physical activity. In our original analysis, we found the further adjustment for energy intake (Model 3) did not substantially change the associations between dietary patterns and the risk of chronic diseases. Therefore, we included two models (Model 1: age, sex, and energy intake; Model 2: full covariates) in the manuscript as suggested by the Reviewer. The results have been updated in Tables S1-S12, and Figures 1-5, S5-S14 as well as the text in the manuscript.

Did BMI influence the associations between dietary patterns and outcomes? The paper mentions BMI in the text but none of the tables/figures include BMI. BMI could be considered a confounder or a mediator.

Response:

We agree with the Reviewer that BMI as an important indicator of metabolic health may influence the associations between dietary patterns and outcomes. In our original analyses, we analyzed obesity (defined by BMI) as a moderator for the association between dietary patterns and incident diseases. As the Reviewer suggests, we have included BMI as a confounder in Model 2. Some associations have been attenuated to be non-significant after the adjustment for BMI. We have updated the results in Tables/Figures.

I might be cautious between a causal connection between diet and psychologic diseases such as depression. There is a link between mental states and poor diets. For e.g., people in a situation/circumstances or personalities prone to stress/anxiety may have poor diets and then over time, some of these people would be more likely to be diagnosed (e.g., with depression).

Response:

We agree with the Reviewer that individuals in a stressful situation may have poor diets, which might have biased the association between diet and psychological diseases. However, we did a sensitivity analysis by excluding individuals who developed psychologic diseases in the first four years of follow-up and found that the results for the associations between dietary patterns and incident psychologic diseases were similar to those in the main analyses. This may reduce the risk of bias that the outcome occurred before the exposure assessment. We have mentioned this as a limitation:

We cannot rule out the potential reverse causation between diet and psychological diseases as people in a situation or personalities prone to stress/anxiety could potentially adopt unhealthy dietary patterns and thus were more likely to be diagnosed with psychological conditions during follow-up.

Minor comments:

Line 137. The scoring method for EDII as shown in Table S4 is different from what it was originally developed. The original score is a weighted sum of 18 food groups (as continuous variables) multiplied by specific weights. It seems the authors have modified EDII. I think the authors need to emphasize this in the methods and explain why they modified it.

Response:

We modified the EDII because it is possible that the EDII score may largely depend on one or two components if individuals consumed too much of these food components. We have added explanations for this in the Methods section:

To avert this scenario, the EDII score could become heavily reliant on just one or several components if individuals consumed too much of these food components, the maximum and minimum scores for individual food items were set at the levels given by previous dietary scores. Scores for the intakes between the maximum and minimum scores were proportionately calculated.

Line 50. Second sentence of the Intro is incomplete.

Response:

We have revised the sentence accordingly:

Meanwhile, ageing is one of the most important risk factors for the development of non-communicable diseases (NCDs), including cardiovascular disease (CVD), diabetes, cancers, and neurodegenerative diseases.

Line 52. How much is the total number of deaths increased due to increasing size of older population?

Response:

We have added this number:

As the percentage of people aged 65 years and older grew from 6.1% in 1990 to 8.8% in 2017, this demographic shift was linked to an additional 12 million global deaths.⁴

Line 184.

- Some dietary patterns include alcohol component. Inclusion of alcohol intake as a covariate might lead to overadjustment.

Response:

We thank the Reviewer for raising this concern. We have excluded alcohol intake as a covariate in the updated analyses.

- The authors need to justify why they include GRS for longevity as a covariate for different diseases. In Table 1, GRS has little variation across dietary score quintiles (as would be expected).

Response:

We have added the justification for why we included GRS for longevity as a covariate:

We adjusted for GRS in the analysis given that it predicted an individual's genetic predisposition to experiencing a lower risk of age-related diseases.

Line 204. I would appreciate a transparent reporting of number of hypotheses being considered.

Response:

We have added the information accordingly:

In our study, we linked four distinct dietary patterns to the risk of 48 different chronic diseases. For these multiple comparisons, Benjamin-Hochberg's procedure was used to control the false discovery rate (FDR) at a 5% level.³⁶

Line 230-235: These numerical values are getting crowded. If you have included them in figures, you don't need to mention all of them.

Response:

As the Reviewer suggests, we have removed the numerical values for other cardiometabolic disorders but CVD.

Line 299" I thought the Mediterranean diet supported higher fish intake, but here it is stated that lower fish intake contributed to the benefit?

Response:

We thank the Reviewer for raising this typo error. It should have been red meat intake. We have revised the sentence as:

The major contributors to the benefits of AMED were higher intakes of whole grains, vegetables, fruits, nuts, legumes, and fish and lower red meat intakes.

Fig1-2. Some associations appeared marginally significant (95% CI includes 1). It's not clear why the authors reported them to be statistically significant even after FDR correction.

Response:

Although these associations with the upper CIs of the HRs are 1 (actually these numbers are 0.996, or 0.997 or such data), the p-values are smaller than 0.05. It is the problem regarding decimal spaces. For those 95% CIs containing 1, we have revised the results to include values with three or more decimal places.

It is not clear how many cases were included for each disease. Does total cancer include non-melanoma skin cancer?

Response:

We have included the number of each disease in figures 1-5, S5-S14.

The total cancer does not include non-melanoma skin cancer. We have added the information in the footnotes of figures 2, 6, S1-S4, S6, and S11, and Tables S1-S12.

The text mentions about average follow-up years for two diseases. I think a supplementary table of median follow-up (and IQR) for each disease will be useful.

Response:

The incidence of dyspepsia as the most incident disease was 9.5% such that the median duration of follow-up exhibited slight variations (ranges from 8.64-8.66 years). That is why we showed the average follow-up duration for the two diseases with the largest difference.

Fig 6. I wonder why the authors only look at components of alternate Mediterranean diet. I can see that AMED was inversely associated with the largest number of diseases in this paper. Did the authors also consider effect size and overall disease burden when interpreting the results?

Response:

The Reviewer is correct that we looked at the components of alternate Mediterranean diet which was inversely associated with the largest number of diseases. Firstly, these components represent the most important dietary factors for health. Secondly, most of the components overlapped with those of other dietary scores including AHEI-2010 and healthy plant-based dietary index.

For the effect size and overall disease burden, we have added the number of event cases and participants as well as incidence for each disease in Figures 1-5, and S5-S14. We have also added the description of the effect size and disease burden in the Results section.

Fig S1-4. I think all results from subgroup analyses should be presented.

Response:

According to the statistical convention, stratified analyses can be conducted if the interaction is significant. Given the multiple comparisons, we just displayed the results with significant interactions after controlling FDR. For those without significant interactions, the associations between dietary patterns and incident diseases were similar across subgroups. We have added more results from subgroups analyses accordingly (new Figures S1-S4).

Line 328. Reference 30 seems to be incorrect.

Response:

We have checked the reference.

Line 342. While some items may overlap between AMED and ketogenic diet, AMED is not ketogenic. Thus, any advantage of the AMED diet is not related to ketogenesis.

Response:

We agree with the Reviewer on this point, and we have revised the explanation for the inverse association between AMED and epilepsy risk:

A diet rich in antioxidants and anti-inflammatory compounds, as found in the Mediterranean diet, may contribute to reducing inflammation and oxidative stress in the brain,^{46,47} which are risk factors for epilepsy. This may partly explain why we found an inverse association between AMED and incident epilepsy.

Some findings related to the AEDII are likely due to the fact that alcohol is in the score (e.g. association with alcohol use disorder, psychoactive substance abuse; weak effect for lung cancer because alcohol drinkers tend to smoke too).

Response:

We have explained this in the Discussion section:

The positive association between AEDII and the risk of alcohol use disorder could be explained by the substantial role of alcohol consumption as a key component of AEDII.

We did not find a significant association between AEDII and the risk of lung cancer.

Reviewer #2 (Remarks to the Author):

The paper has well designed; analyses were appropriate. More evidence related healthy diet as a protective factor.

Response:

We thank the Reviewer for his or her positive comments on our manuscript.

However, these scores are based on different criteria, and it is unclear which score best predicts chronic disease risk.

Response:

We agree with the Reviewer that our statement was not clear regarding the best score for the prediction of chronic diseases. AMED was linked to the largest number of chronic diseases and yielded the lowest risk for most chronic diseases. We have added the information in the first paragraph of the Discussion section:

In this large cohort study, we found a higher AMED score was associated with a lower risk of 32 (all 8 CMDs, 3 out of 10 types of cancers, 7 out of 10 psychological/neurological disorders, 5 out of 6 digestive disorders, and 9 out of 14 other chronic diseases) out of 48 chronic diseases. AHEI-2010 was inversely associated with the risk of 29 chronic diseases (7 CMDs, 4 cancers, 5 psychological/neurological disorders, 5 digestive disorders, and 8 other chronic diseases). A higher HPDI score was associated with a reduced risk of 23 chronic diseases (6 CMDs, 4 cancers, 4 psychological/neurological disorders, 5 digestive disorders, and 4 other chronic diseases). No positive associations between AMED, AHEI-2010, and HPDI and the risk of any

chronic disease were observed. AEDII was inversely associated with the risk of 14 chronic diseases and positively associated with the risk of two chronic conditions (alcohol use disorder, psychoactive substance abuse). AHEI-2010 demonstrated the lowest risk for alcohol use disorder and psychoactive substance abuse, AEDII showed the lowest risk for diabetes and thyroid disorders, while AMED yielded the lowest risk for many other chronic diseases (CVD, cancer, COPD, CKD, chronic liver disease, psychological/neurological disorders, and digestive disorders).

There is a lot of information which was well summarised, but we lost the importance of diet in each disease, including all in the same paper.

Response:

We thank the Reviewer for raising this important concern. We agree with the Reviewer that there are both pros and cons to investigating a broad range of diseases. We have mentioned this as a limitation:

Sixthly, investigating a broad range of chronic diseases offers certain benefits, but is also limited by narrowing the focus to a specific disease (discussion of the mechanisms).

Meanwhile, we have added more statements regarding which score yielded the lowest risk for each disease.

Analyses were adjusted by confounding, but maybe moderator analysis by obesity, education, or other factors specific for each disease, which would be interesting too and it's lost.

Response:

We agree with the Reviewer that it is interesting to do moderation analysis by important factors including obesity, education, hypertension, diabetes, and dyslipidemia. We have added the results for moderation analyses in Figures S1-S4:

The inverse association between AMED and the incidence of irritable bowel syndrome, osteoporosis, dyspepsia, and cataract was stronger among individuals with hypertension/dyslipidemia (Figure S1). An AEDII score was more predictive of diabetes/CKD among younger than in older individuals (Figure S2). Age and metabolic disorders were significant moderators for the association between AHEI-2010 and the risk of cancer or cataract. The inverse association between AHEI-2010 and incident dyspepsia was stronger among individuals with lower education (Figure S3). The association between HPDI score and incident hypertension was stronger in younger than older individuals (Figure S4).

REVIEWERS' COMMENTS

Reviewer #1 (Remarks to the Author):

The authors responded adequately to all concerns raised. Thank you.

Reviewer #2 (Remarks to the Author):

No further comments